



# Lithosphere tearing along STEP faults and synkinematic formation of lherzolite and wehrlite in the shallow subcontinental mantle

Károly Hidas[1,2*], Carlos J. Garrido[1], Guillermo Booth-Rea[2], Claudio Marchesi[1,3], Jean-Louis Bodinier[4],
Jean-Marie Dautria[4], Amina Louni-Hacini[5], Abla Azzouni-Sekkal[5,6]

1 Instituto Andaluz de Ciencias de la Tierra, CSIC & Universidad de Granada, Av. de las Palmeras 4, 18100 Armilla (Granada), Spain

2 Departamento de Geodinámica, Facultad de Ciencias, Universidad de Granada, Campus de Fuentenueva s/n, 18071 Granada, Spain

3 Departamento de Mineralogía y Petrología, Facultad de Ciencias, Universidad de Granada, Campus de Fuentenueva s/n, 18071 Granada, Spain

4 Géosciences Montpellier, UMR 5243, CNRS & Université de Montpellier, Place E. Bataillon, 34095 Montpellier, France

5 Faculté des Sciences de la Terre, de Géographie et de l'Aménagement du Territoire - Université des Sciences et de la Technologie Houari Boumédiène - Laboratoire de Métallogénie et Magmatisme de l'Algérie, 32, El Alia, 16111, Bab Ezzouar, Algiers, Algeria

6 Faculté des Sciences de la Nature et de la Vie et des Sciences de la Terre et de l'Univers, Université Abou Bekr Belkaïd, BP. 119, 13000, Tlemcen, Algeria

*Correspondence to*: K. Hidas (karoly.hidas@csic.es)

**Abstract.** Subduction-Transform Edge Propagator (STEP) faults are the locus of continual lithospheric tearing at slab edges, resulting in sharp changes in the lithospheric and crustal thickness and triggering lateral and/or near-vertical mantle flow. However, the mechanisms at the lithospheric mantle scale are still poorly understood. Here, we present the microstructural study of olivine-rich lherzolite, harzburgite and wehrlite mantle xenoliths from the Oran volcanic field (Tell Atlas, NW Algeria). This alkali volcanic field occurs along a major STEP fault responsible for the Miocene westward slab retreat in the westernmost Mediterranean. Mantle xenoliths provide a unique opportunity to investigate the microstructures in the mantle section of a STEP fault system.

The microstructures of mantle xenoliths show a variable grain size ranging from coarse granular to fine-grained equigranular textures uncorrelated with modal variations. The major element composition of the mantle peridotites provides temperature estimates in a wide range (790–1165 °C) but in general, the coarse-grained and fine-grained peridotites suggest deeper and shallower provenance depth, respectively. Olivine grain size in the fine-grained peridotites depends on the size and volume fraction of the pyroxene grains, which is consistent with pinning of olivine grain growth by pyroxenes as second phase





particles. In the coarse-grained peridotites, well-developed olivine crystal preferred orientation (CPO) is characterized by orthorhombic and [100]-fiber symmetries, and orthopyroxene has a coherent CPO with that of olivine, suggesting their coeval deformation by dislocation creep at high-temperature. In the fine-grained microstructures, along with the weakening of the fabric strength, olivine CPO symmetry exhibits a shift towards [010]-fiber and the [010]- and [001]-axes of orthopyroxene are

5 generally distributed subparallel to those of olivine. These data are consistent with deformation of olivine in the presence of low amounts of melts and the precipitation of orthopyroxenes from a melt phase. The bulk CPO of clinopyroxene mimics that of orthopyroxene via a topotaxial relationship of the two pyroxenes. This observation points to a melt-related origin of most clinopyroxenes in the Oran mantle xenoliths.

The textural and geochemical record of the peridotites are consistent with interaction of a refractory harzburgite protolith

with a high-Mg# melt at depth (resulting in the formation of coarse-grained clinopyroxene-rich lherzolite and wehrlite), and with a low-Mg# evolved melt in the shallow subcontinental lithospheric mantle (forming fine-grained harzburgite). We propose that pervasive melt-peridotite reaction —promoted by lateral and/or near-vertical mantle flow associated with lithospheric tearing— resulted in the synkinematic crystallization of secondary lherzolite and wehrlite and played a key effect on grain size reduction during the operation of the Rif-Tell STEP fault. Melt-rock reaction and secondary formation of

lherzolite and wehrlite may be widespread in other STEP fault systems worldwide.

# 1 Introduction

Slab edges are necessary ingredients of plate tectonics, and vertical tearing along the lithosphere at the termination of subduction trenches is a geometric consequence that enables the subduction to continue (e.g., Wilson, 1965;Millen and Hamburger, 1998;Govers and Wortel, 2005;Nijholt and Govers, 2015). A Subduction-Transform Edge Propagator (STEP)

fault is the locus of continual lithospheric tearing at slab edges, which allows subduction of one part of a tectonic plate, while the juxtaposed part remains at the surface (Govers and Wortel, 2005;Nijholt and Govers, 2015). The evolution of the subduction zone along STEP faults results in sharp changes in the lithospheric and crustal thickness, which trigger lateral and/or near-vertical mantle flow associated with lithospheric tearing (e.g., Mancilla et al., 2015;Menant et al., 2016). Upwelling melts that form in this process are likely channelized by the translithospheric STEP faults and may lead to intense melt-rock

reaction in the shallow subcontinental lithospheric mantle (SCLM).

In terms of modal composition, melt-rock reaction processes that occur at shallow mantle depths result in a lithological variation trending from refractory peridotites towards lherzolites with a high clinopyroxene/orthopyroxene modal ratio, eventually forming wehrlite (e.g., Peslier et al., 2002;Parkinson et al., 2003;Ionov et al., 2005;Soustelle et al., 2009;Lambart et al., 2012;Bodinier and Godard, 2014;Varas-Reus et al., 2016;Marchesi et al., 2017). The microstructure of such olivine- and

30 clinopyroxene-rich SCLM peridotites usually indicates syn- to post-kinematic melt-rock reactions (e.g., Tommasi et al., 2004;Tommasi et al., 2008;Morales and Tommasi, 2011;Zaffarana et al., 2014;Kourim et al., 2015). While wehrlite and related peridotite lithologies are frequent in the shallow SCLM, the impact of STEP fault in their formation in subduction environments





is poorly understood. The western Mediterranean, however, provides a natural laboratory to study the relationship between lithosphere tearing along a STEP fault and its relationship with wehrlitization of the shallow SCLM.

The Betic-Rif orogen has been proposed to form during the westward retreat of the western Mediterranean subduction system (Lonergan and White, 1997;Gutscher et al., 2002;Booth-Rea et al., 2007). Beneath the Betics, a sharp and prominent lithospheric step at the termination of the Iberian lithosphere is interpreted as a near-vertical STEP fault structure associated with lateral lithospheric tearing (Mancilla et al., 2015;Mancilla et al., 2018). Deformation pattern and seismic anisotropy signal of mantle xenoliths from the eastern Betics indicate steeply dipping foliation and subhorizontal lineations in the shallow SCLM, which are consistent with WSW tearing of the subducted south Iberian margin lithosphere along the STEP fault (Hidas et al., 2016a). Moreover, in the same volcanic field, wehrlite lithologies are interpreted as an interaction with $SiO_2$-understaturated magmas, similar to the alkali basalts that host the mantle xenoliths (Marchesi et al., 2017). The Oran volcanic field is situated on the northern margin of the African plate and its SCLM records processes related to slab edge delamination and asthenospheric mantle upwelling coupled to the rapid westward retreat of the subduction arc system along the Rif-Tell STEP fault, predating the latemost Miocene development of the Betic STEP fault in Southeastern Betics (Booth-Rea et al., 2007;Garcia-Castellanos and Villasenor, 2011;Platt et al., 2013;van Hinsbergen et al., 2014;Chertova et al., 2014a;Chertova et al., 2014b).

The Plio-Pleistocene basaltic volcanism in the Oran volcanic field transported a large number of mantle xenoliths to the surface, mostly in the Aïn Témouchent volcanic complex. Earlier works on the peridotite suite have shown that the SCLM records metasomatic events in the shallow spinel- and plagioclase-facies due to channeling of reactive alkaline magmas in ductile shear zones (Zerka et al., 2002;Lahmer et al., 2018). However, due to the lack of microstructural data, the reconstruction of deformation mechanisms and the relationship between the melt percolation and deformation along the Rif-Tell STEP fault have remained unexplored. In this study, we provide new microstructural and major element geochemical data on a representative set of mantle peridotites from various localities of the Oran volcanic field. We aim at constraining the physico-chemical processes coupled to the lithospheric tearing at a STEP fault in the context of the western Mediterranean geodynamic evolution along the northern margin of the African plate.

## 2 Geologic setting and sampling strategy

The Betic-Rif-Tell orogen in the westernmost Mediterranean (Fig. 1a) was formed during the Miocene collision between the westward-migrating Alborán terrain and the south Iberian and the Maghrebian passive margins in the context of Africa-Iberia convergence (Jolivet and Faccenna, 2000;Faccenna et al., 2004;Spakman and Wortel, 2004;Booth-Rea et al., 2007). This geodynamic evolution resulted in extension due to slab rollback, slab detachment, mantle lithosphere delamination and convective lithospheric thinning (e.g., Platt et al., 2013 and references therein). 3-D tomographic imaging of the mantle structure reveals a steeply east-dipping slab under the Gibraltar strait with a slab morphology that extends from under the Rif towards the north and then curves eastward under the Betics (e.g., Spakman and Wortel, 2004;Gutscher et al., 2012;Bezada et



al., 2013;van Hinsbergen et al., 2014). These works also show a detached slab beneath the Kabylides in northeast Algeria (Spakman and Wortel, 2004;Faccenna et al., 2014;van Hinsbergen et al., 2014), which is interpreted as a part of the Algerian-Tunisian slab separated from the Betic-Rif slab system by STEP faults (Booth-Rea et al., 2018a). The present-day Betic-Rif slab morphology is consistent with the evolution of an initially NW dipping subduction zone situated southeast of the Balearic

islands in the Oligocene that retreated towards the south and later to the west, after approximately 90º clockwise rotation, during the Miocene to reach its current position beneath the Gibraltar strait (e.g., Royden, 1993;Lonergan and White, 1997;Rosenbaum et al., 2002;Spakman and Wortel, 2004;Chertova et al., 2014a;Chertova et al., 2014b;van Hinsbergen et al., 2014). Slab tearing and detachment in the southern part of the slab is associated with the rapid westward retreat of the subduction system (e.g., Roure et al., 2012;Abbassene et al., 2016;Chazot et al., 2017) that created a STEP fault parallel to the

N-African margin (Mauffret et al., 2004;Medaouri et al., 2014;Chertova et al., 2014a and references therein).

    In this tectonic context, subduction-related to intraplate anorogenic magmatism occurred at various places at different geodynamic stages. Neogene to Quaternary volcanism occurred mainly in SE Iberia, the Maghrebian margins and the present-day Alborán Sea basin (Fig. 1a) (Hoernle et al., 1999;Turner et al., 1999;Duggen et al., 2004;Duggen et al., 2005). In the Early to Middle Miocene, the volcanic activity was broadly coeval with extension and subsidence of the Alborán Sea Basin and

orogenesis in the surrounding Betic-Rif-Tell mountain belt (Lonergan and White, 1997;Comas et al., 1999;Jolivet and Faccenna, 2000;Booth-Rea et al., 2007). Middle to late Miocene tholeiitic through calc-alkaline volcanic rocks typical of subduction arc magmatism crop out across the central and eastern Alborán Sea Basin (Fig. 1a), in the Alborán Island, and in coastal volcanic complexes at the margins of the eastern Betics and the Maghrebides (Fig. 1a) (Turner et al., 1999;Duggen et al., 2004; Duggen et al., 2005 and references therein). Actually, the eastern Alborán basin basement is formed by a submerged

well-developed volcanic arc (Booth-Rea et al., 2018b;Gómez de la Peña et al., 2018). Meanwhile, at the W-Maghrebian and SE-Iberian continental margins, this late Miocene magmatism was coeval to extensional collapse and thinning of the previously thickened orogenic crust (e.g., Booth-Rea et al., 2012;Giaconia et al., 2014;Jabaloy-Sánchez et al., 2015;Azdimousa et al., 2019), and loss of its SCLM root (Duggen et al., 2003;Mancilla et al., 2015). This process is defined as edge delamination by Duggen et al. (2003). Late Miocene to Pliocene topographic uplift and coeval Si-K-rich magmatism and Quaternary alkali

basalts in SE-Iberia and the Maghrebian margins have been related to asthenospheric mantle upwelling after late Miocene edge delamination (Duggen et al., 2003;Duggen et al., 2005;Duggen et al., 2008) and/or earlier slab break-off in the Algerian-Rifean margin (Spakman and Wortel, 2004;Chazot et al., 2017). Mantle xenoliths in Plio-Quaternary alkali basalts have been interpreted to have registered part of this tectonic evolution in the mantle underlying the Betics, preserving the originally subvertical tectonic foliation (Hidas et al., 2016a) probably related to mantle flow during slab tearing along the Betic STEP

(Mancilla et al., 2015;Mancilla et al., 2018).

    The Oran volcanic field in NW Algeria is dominated by Miocene to Plio-Pleistocene calc-alkaline and alkaline volcanic rocks (Fig. 1b) (Louni-Hacini et al., 1995;Coulon et al., 2002;Bendoukha et al., 2009). The volcanic rocks crosscut and overlie the Tell sedimentary nappes overthrust during the Miocene on the northern part of the African plate (Coulon et al., 2002). The alkaline lavas contain a large number of mantle xenoliths (Zerka et al., 2002;Lahmer et al., 2018 and references therein). In



this study, we present mantle xenoliths from two localities. Samples with label SOU have been collected in the 2.1 Ma lava flows at the Oued Abdellah beach of Souahlia (35° 5'45.58"N / 1°52'33.23"W), whereas those with labels DZ (Djebel Dzioua; 35°17'21.36"N / 1°14'43.71"W), GU (Djebel Gueriane; 35°11'28.80"N / 1°12'27.00"W) and HAM (Hammar E'Zohra; 35°15'42.81"N / 1°8'41.65"W) are from the 1.3-0.8 Ma basaltic flows in the Aïn Témouchent massif (Fig. 1b). The 25 selected

samples (7 DZ, 4 GU, 11 HAM and 3 SOU; Table 1) represent all textural and lithological varieties that occurred in a given outcrop.

## 3 Analytical methods

### 3.1 Sample preparation

Mantle xenoliths from the Oran volcanic field are fresh but their impregnation by fast-acting adhesives and/or epoxy under

vacuum was usually necessary before and during thin section preparation. Most xenoliths display no penetrative macroscopic foliation and/or lineation, thus thin sections were cut at random orientations. In the few samples where foliation and lineation are observable, thin sections were cut perpendicular to foliation and parallel to lineation; the latter is defined by elongated spinels and orthopyroxene porphyroclasts (stretching lineation). Standard petrographic thin sections were grinded to a thickness of ca. 150 microns and were polished using diamond paste with 3 and 1 micron grain sizes. Final surface of the thin

sections was achieved after 45 minutes chemical and mechanical polishing using Buehler MasterMet colloidal silica polishing suspension.

### 3.2 Microstructural analyses

For electron backscatter diffraction (EBSD) analyses, the uncoated thin sections were mounted with conductive carbon tape to reduce charging. Analyses were carried out at high vacuum (in the range of $10^{-5}$ mbar) using a Zeiss EVO MA 15 SEM

equipped with Oxford Instruments Nordlys Nano EBSD detector and AZtec v. 3.1 data acquisition software (Oxford/HKL) at the Instituto Andaluz de Ciencias de la Tierra (IACT, Armilla, Granada, Spain). Instrumental settings were 17 kV acceleration voltage, 24 mm working distance, probe current of 0.5-1.0 nA, using Agar Scientific tungsten filament as a source of electrons. Acquisition conditions in the EBSD software module were 4×4 binning and low (0) gain with grid steps varied between 17 and 40 µm (depending on the grain size), covering most of the sample surface. All major constituent minerals of the rocks

were included in the phase list and the percentage of indexed points in the raw maps typically exceeded 75%. In some xenoliths (samples DZ-005, GU-003, HAM-004, HAM-006, HAM-016), the interstitial adhesive used for impregnation has resulted in local charging effect of the thin section surface, reducing the EBSD indexing rate to 40-60%. The obtained data, however, are representative for the entire thin section, and EBSD data from these samples have been included in the database. Post-acquisition data treatment to clean raw maps was carried out following the procedure presented in Soustelle et al. (2010), using

the built-in functions of the Oxford/HKL Channel 5 software package.





For further data processing and calculations —carried out on cleaned EBSD datasets— we used the built-in functions of the free MTEX 4.5.2 Matlab toolbox (https://mtex-toolbox.github.io; Hielscher and Schaeben, 2008;Bachmann et al., 2010) (Table S1). Grain boundaries are defined at 12° minimum misorientation between neighboring pixels (segmentation angle); the correctness of grain definition have been checked in optical microscope. Low misorientation boundaries in the range of 2-

12° are considered as subgrain boundaries. From these data, we calculated the total length of the subgrain boundaries per mineral phase, and in each sample, we estimated the subgrain density for a given mineral by dividing the subgrain boundary length with the total area of the corresponding mineral phase (data expressed as 1/mm). Several methods have been proposed for strain analysis from EBSD data (see Wright et al., 2011 for a review). In the present study, we characterize intragranular misorientations by the misorientation relative to the mean orientation of a grain (Mis2Mean), which is the deviation in

orientation at a measurement point from the mean orientation of the grain to which the point belongs. The spread in orientations within a grain is characterized by the grain orientation spread (GOS), which is the average of the Mis2Mean over the grain. Sharp gradients in misorientations, such as subgrain boundaries, are best visualized by calculating second nearest-neighbor Kernel Average Misorientations (2nd order KAM, hereafter KAM2) maps. The KAM2 is the deviation in orientation of a measurement pixel from the average orientation of its nearest 12 neighbors. Here it is calculated with a cut-off value of 12°

(i.e., segmentation angle used for defining a high angle boundary). To compare samples to each other, in the analyzed surface we calculate simple averages of KAM2 and Mis2Mean data, and area-weighted averages for GOS and descriptive parameters of grain morphology (e.g., shape, size) to reduce the effect of strain-free small grains. To characterize the symmetry of olivine CPO, we calculated the BA-index (Mainprice et al., 2014) based on the equation of BA-index=$0.5\times[2-[P_{010}/(G_{010}+P_{010})]-[G_{100}/(G_{100}+P_{100})]]$. This parameter considers the point (P) and girdle (G) distribution of [010] and [100] crystallographic axes.

For each axis these distributions are calculated from the orientation tensor and its three eigenvalues $\lambda_1$, $\lambda_2$, $\lambda_3$ (where $\lambda_1 \geq \lambda_2 \geq \lambda_3$ and $\lambda_1+\lambda_2+\lambda_3=1$; Vollmer, 1990) as P= $\lambda_1-\lambda_2$ and G=$2\times(\lambda_2-\lambda_3)$. For a perfect [010]-fiber CPO, the P and G values for [100] and [010] are 0, 1 and 1, 0, respectively, and the BA-index is 0. In the other end-member case, a perfect [100]-fiber CPO the P and G values for [100] and [010] are 1, 0 and 0, 1, respectively, hence the BA-index is 1. In a perfect orthorhombic fabric, for both [100] and [010] P and G values are 1 and 0, respectively, thus the BA-index is 0.5. Note that this index only specifies CPO

symmetry, but not the position of crystallographic axes with respect to structural elements in the rock (lineation and foliation). Finally, we calculate the strength of CPO (i.e., fabric strength) using the misorientation index (M-index) developed by Skemer et al. (2005). The M-index is 0 for random fabric and 1 for a single crystal; in highly deformed samples its empirical value is approximately 0.45.

For oriented thin sections, the crystal-preferred orientation (CPO) is shown in the structural reference frame with lineation

and pole to the foliation at the E-W and N-S directions of the pole figures, respectively. For randomly cut thin sections, measured CPO were rotated into a common frame of reference, in which the maximum concentration of olivine [100] and [010] axes are parallel to the E-W and N-S directions of the pole figures, respectively. Note that this rotation does not affect neither the intensity nor the symmetry of the CPO, and it does not influence the relative orientation of crystallographic axes of different mineral phases in a given sample. To avoid overrepresentation of large crystals, we plot CPO for average grain



orientations ('one point per grain') instead of individual measurements using the careware software package of David Mainprice (ftp://www.gm.univ-montp2.fr/mainprice//CareWare_Unicef_Programs/).

## 3.3 Mineral chemistry

Quantitative analyses of major element compositions of minerals were carried out on a CAMECA SX 100 electron microprobe
(EMPA) at the Centro de Instrumentación Científica (CIC) of the University of Granada (Spain). Accelerating voltage was 15–20 kV, with a sample current of 10–15 nA and a beam diameter of 5 μm. Counting time for each element was 10–30 s. Natural and synthetic silicate and oxide standards were used for calibration and the ZAF correction. The results of major element analyses of minerals are listed in Table S1.

## 4 Petrography

### 4.1 Lithologies

The studied mantle xenoliths from Oran are typically 5-7 cm in diameter and the majority of the samples are from the spinel facies. Rare plagioclase (up to 0.8 %) occurs mostly at the HAM locality in the Aïn Témouchent massif (Table 1). The majority of the mantle xenoliths show a modal compositional trend from fertile lherzolite to olivine-rich harzburgite, and the refractory lithologies —such as olivine-rich lherzolite and harzburgite— are the most abundant (Fig. 2a). There is a population, however,
that reflects anomalously high clinopyroxene contents and can be classified as clinopyroxene-rich lherzolite and wehrlite (<10 % of the samples) (Fig. 2a).

Irrespective of the locality and the lithology, hydrous phases (mostly amphibole) are also present in low modal amounts (<0.6 %) (Table 1). Indications for the presence of interstitial glass has also been found, although the current state of preservations of the xenoliths did not allow their further investigation. These petrographic features, the observed lithologies
and their abundances are nevertheless consistent with earlier reports from the same volcanic field (e.g., Zerka et al., 2002;Lahmer et al., 2018).

### 4.2 Textures and microstructures

The most striking textural feature of the studied xenoliths from Oran is the gradual decrease and homogenization of the grain size from 1.5-2.0 mm to 0.3 mm due to the increasing amount of fine-grained, dispersed pyroxenes ± olivine in the fabric (Fig.
2b, Table 1). Based on the overall grain size and the bimodal or homogeneous grain size distribution of the silicate phases, we group the xenoliths into four textural types. The two endmembers are characterized by rather homogeneous grain size distribution accompanied with several millimeter large olivine and orthopyroxene in the coarse granular (6 samples; 24%), and submillimetric olivine and pyroxenes in the equigranular texture (9 samples; 36%) (Fig. 2b). Probably the latter texture type was considered as a mylonite in earlier works from the Oran (e.g., Zerka et al., 2002). The transition is marked by textures
that contain large orthopyroxene and, rarely, clinopyroxene crystals surrounded by a significantly finer-grained matrix





composed of olivine and pyroxenes. Based on the average grain size of olivine in these transitional textures, we distinguish coarse-grained porphyroclastic (olivine >0.8 mm; 8 samples, 32%) and fine-grained porphyroclastic texture types (olivine <0.8 mm; 2 samples, 8%) (Fig. 2b). Hereafter we address the coarse granular and the coarse-grained porphyroclastic textures as coarse-grained, and the fine-grained porphyroclastic and equigranular textures as fine-grained rocks (Fig. 2b).

In the coarse-grained rocks, large olivines show moderate intracrystalline deformation features. They contain widely spaced subgrain boundaries perpendicular to the weak elongation direction of the minerals and have an undulose extinction in crossed-polarized light optical microscope (Fig. 3a-c). The grain boundaries of large olivine grains are typically lobate but dynamic recrystallization into strain-free neoblasts is rarely observed (Fig. 3a-b). Rather than a mantle around porphyroclasts —typical for dynamic recrystallization— the fine-grained olivine fraction occurs locally along interstitial patches intermixed with fine-

grained ortho- or clinopyroxene (Fig. 3b). When olivine is intermixed with pyroxenes, its grain size is significantly smaller than in pyroxene-free areas (Fig. 3b). In the fine-grained rocks, olivine contains negligible intragranular misorientations and generally lacks subgrain boundaries (Fig. 3d-f). Grains are often polygonal with straight or gently curved grain boundaries meeting at 120° triple junctions, particularly in the equigranular texture (Fig. 3e-f).

    Pyroxenes are present in two remarkably different textural positions. On the one hand, orthopyroxene can form

porphyroclasts up to 0.5 cm in diameter slightly elongated in shape (Fig. 3b), with aspect ratios up to 4:1. These porphyroclasts frequently contain exsolution lamellae of clinopyroxene in orthopyroxene (Fig. 3a), and have strongly curved, sutured grain boundaries (Fig. 3b) that contain fine-grained olivine in their embayments (Fig. 3b,e). Dynamic recrystallization into strain-free neoblasts is present but is not common. Clinopyroxene may also form large crystals that are not porphyroclasts but aggregates of several millimetric crystals in elongated patches in the plane of the foliation (Fig. S1). Large pyroxenes are rarely

observed in the equigranular texture. On the other hand, both pyroxenes occur as a fine-grained (<100 microns in diameter) and strain-free mineral fraction, usually dispersed in the rocks without showing textural preference to porphyroclasts. These small pyroxenes are highly irregular in shape with convex grain boundaries and form diffuse bands along olivine-olivine grain boundaries or have cusp-like termination at triple junctions (Fig. 3e). Small pyroxenes are rare in the coarse-grained rocks and they are the most abundant in the equigranular texture. Small orthopyroxene and clinopyroxene are generally present with a

proportion ranging from 80:20 to 60:40 (Fig. 3e), respectively, but they typically occur in monophase patches rather than showing phase mixing. In some clinopyroxene-rich depleted lherzolite (e.g., DZ-003), the fine-grained clinopyroxene occur as a rim on orthopyroxene porphyroclasts locally showing reaction microstructure at the phase boundary and identical crystallographic orientation of the two pyroxenes (Fig. 3b). These clinopyroxenes also host a large amount of primary fluid/melt inclusions.

Dark brown, millimetric spinel crystals are typically elongated in the plane of the foliation parallel to the lineation, or less commonly, they appear with holly-leaf structures filling interstices between silicates. A minor, fine-grained spinel fraction (<200 microns in diameter) occurs together with fine-grained pyroxenes at the grain boundaries of olivine. Plagioclase, if present, is fine-grained (<200 microns in diameter) and forms polygonal crystals surrounding large spinel. Rarely, it is observed



as an interstitial phase intermixed with fine-grained pyroxenes. Amphibole, if present, shows no textural variability and occurs as fine-grained (50-100 microns in diameter) interstitial crystals.

## 4.3 Crystal Preferred Orientation (CPO)

In all xenoliths from Oran, olivine displays a well-developed CPO with variable intensities and pattern symmetries in the

function of texture. Most coarse-grained rocks are characterized by orthorhombic and [100]-fiber olivine symmetries (BA-indices of 0.5-0.9) and moderate olivine fabric strength (M-indices of 0.17-0.37) (Fig. 4). Although olivine CPO in the majority of the equigranular samples can be also classified as orthorhombic, we observe a minor yet characteristic shift towards the field of [010]-fiber pattern (BA-indices <0.45; Fig. 4). The fabric strength of these latter samples (M-indices 0.03-0.18) is also weaker than that of the coarse-grained rocks. The fine-grained porphyroclastic samples show [100]-fiber CPO symmetries

similar to the coarse-grained rocks but they are accompanied with CPO intensities typical for equigranular textures (M-index <0.12; Fig. 4). Finally, the wehrlite from Souahlia is characterized by a CPO symmetry similar to the equigranular samples (BA-index of 0.39) but with somewhat stronger fabric (M-index of 0.22) (Fig. 4). In the oriented thin sections, the maximum density of [100] axes is subparallel to the stretching lineation and the [010] axes show a maximum normal to the plane of the foliation. In the [100]-fiber CPO symmetries, typical for many coarse-grained xenoliths, the dispersion of [010]-axes form a

girdle in a plane normal to the [100] maximum and the [001]-axes either form a weak girdle parallel to the distribution of [010] axes or are at high angle to the lineation in the plane of the foliation (Fig. 5a). In the orthorhombic CPO symmetries that show tendency towards [010]-fiber pattern, typical for some fine-grained xenoliths, we observe a strong maximum of [010] perpendicular to the foliation and a very weak girdle of [100] and [001] axes in the plane of the foliation (Fig. 5b). Irrespective of the CPO symmetry, the concentration of [100] axes is always stronger than that of the [001] (Fig. S1). The observed olivine

CPO patterns in the Oran xenoliths are the most abundant ones among those reported from mantle xenoliths and massive peridotites from different geodynamic environments worldwide (Tommasi et al., 2000;Tommasi and Vauchez, 2015).

The CPO of orthopyroxene in most samples is more dispersed than that of olivine, and the fabric strength and the alignment of crystallographic axes with respect to structural elements show a strong correlation with texture type. In the coarse-grained xenoliths, the [001]-axes of orthopyroxene are subparallel to the [100]-axes of olivine, hence the stretching lineation (Fig. 5a).

This is also confirmed on elongated porphyroclasts, where [001]-axes are parallel to the elongation direction of the grains. In addition, the [100]-axes of orthopyroxenes also display weak alignment with the [010]-axes of olivine and they are distributed at the pole of the foliation in the oriented thin sections. In the coarse-grained rocks, orthopyroxene has a moderate fabric strength (M-index of 0.13-0.36), which is positively correlated to that of olivine (Table 1). In the fine-grained rocks — particularly in the equigranular samples— the [010]- and [001]-axes of orthopyroxene are generally distributed subparallel to

the [010]- and [001]-axes of olivine, respectively (e.g., samples HAM-007 and HAM-018 in Fig. 5b). The fabric strength of orthopyroxene is also weak in these samples with M-indices ranging from 0.01 to 0.08 (Table 1).

The crystallographic axes of clinopyroxene are typically distributed subparallel to that of orthopyroxene (Fig. 5), which has also been confirmed locally on a grain-by-grain basis earlier in Fig. 3b. Note that significant misindexing of these two



phases have not been observed in the EBSD maps. The fabric of clinopyroxene is typically weaker than that of orthopyroxene, irrespective of the texture type (Fig. 5) and —apart from three coarse-grained samples— the M-index of clinopyroxene is ranging from 0.01-0.11 (Table 1).

To characterize the rotation axes accommodating low-angle (2–15°) misorientations in olivine and orthopyroxene, we made subsets of large (>400 µm equivalent diameter) and small grains (<400 µm equivalent diameter) in mantle xenoliths representative of each textural group (Fig. 6). In olivine, rotation axes usually show <0vw> orientations irrespective of the grain size but maxima around [010] become stronger with grain refinement and low-angle misorientation are dominantly accommodated around this axis in the small grain fraction of equigranular xenoliths (Fig. 6a). In orthopyroxene, low-angle misorientations are accommodated around [010] axes in large grains but in the small grains rotation around [001] also appears
and sometimes becomes dominant, particularly in fine-grained xenoliths (Fig. 6b).

## 5 Mineral chemistry and geothermometry

### 5.1 Mineral major element composition

The results of EPMA analyses are provided in supplementary Table S1. The average olivine Mg# [Mg#=$100\times$Mg/(Mg+Fe$^{2+}$), in atomic per formula units] in the Oran xenoliths typically plots in the range of 89.4-90.8 regardless of the texture type or
lithology, except for an equigranular, clinopyroxene-rich lherzolite HAM-018 where it is anomalously low (86.1, Fig. 7a). The average values show, however, significant standard variations particularly in the fine-grained rocks and in the wehrlites (Fig. 7a). The average NiO content in olivine typically ranges from 0.34 wt.% to 0.42 wt.% without correlation to texture type, but lower values are more often in wehrlite and clinopyroxene-rich lherzolite (Table S1). We observe no significant differences in major element composition of olivine cores and rims (Table S1).

The average spinel Cr# [Cr#=$100\times$Cr/(Cr+Al), in atomic per formula units] and Mg# encompass a wide range from 9.0-55.0 and 58.7-78.1, respectively, and show no correlation to texture type or lithology. We detect no systematic differences between cores and rims either, but in some samples, the cores are up to 5 wt.% richer in Al$_2$O$_3$ and poorer in Cr$_2$O$_3$ than the corresponding rims (Table S1). Covariation between olivine Mg# and spinel Cr# is normally observed in spinel peridotites from different tectonic settings (Arai, 1994) . Most xenoliths from Oran plot within the field of the olivine-spinel mantle array,
and are distributed along the typical partial melting trend of a fertile mantle source, being residues of relatively high degrees of partial melting (10-25%; Fig. 7a). The only exception is sample HAM-018, which plots to the right of this field due to its low olivine Mg# (Fig. 7a), similarly to wehrlites from the eastern Betics (e.g., Marchesi et al., 2017), as well as strongly metasomatized peridotites and ultramafic cumulates worldwide (Arai, 1994).

The average orthopyroxene Mg# ranges from 89.1 to 91.9, except for the sample HAM-018, where we calculate values
down to 86.1 (Table S1). The average Al$_2$O$_3$ (2.2-5.9 wt.%) and the CaO (0.5-2.5 wt.%) in orthopyroxene are not correlated to texture or lithology but, in general, the concentrations tend to be lower in the fine-grained rocks and towards the mineral rims.





Regardless of texture and lithology, the average clinopyroxene Mg# spans from 88.9 to 93.6, and the $Cr_2O_3$ and $Na_2O$ contents are typically in the range of 0.82-1.3 wt.% and 0.92-1.59 wt.%, respectively (Table S1). The fine-grained rocks tend to have slightly higher Mg# than the coarse-grained rocks, but the ranges overlap. The only exception is the fine-grained sample HAM-018 that has an anomalously low Mg# (86.8-88.0), and $Cr_2O_3$ (0.42-0.77 wt.%) and $Na_2O$ (0.72-0.94 wt.%)

contents (Table S1). We have not observed any systematic intragranular variation in the major element composition of clinopyroxene.

## 5.2 Geothermometry

We calculated the two-pyroxene solvus $T_{Cpx-Opx}$ (Brey and Köhler, 1990) and the Cr-Al-orthopyroxene $T_{Cr-Al-Opx}$ (Witt-Eickschen and Seck, 1991) geothermometers for the core and the rim compositions (Table 1; Fig. 7b). The estimated

uncertainty given by the authors is ±16 °C for both methods. In the SOU-003 wehrlite sample, estimating the temperature was not possible due to the lack of geothermometric formulation in clinopyroxene-olivine assemblages (Nimis and Grütter, 2010).

We obtained a wide range of temperatures (790-1165 °C and 850-1090 °C from cores and rims, respectively) and the results of the geothermometers are usually strikingly different in the same sample. In general, the rims provide lower temperature than the corresponding cores using the $T_{Cr-Al-Opx}$ method, whereas they record similar or slightly higher temperatures than the

cores by the two-pyroxene geothermometer (Fig. 7b). We observe a weak match of the two geothermometric methods only in a few coarse-grained rocks, typically in the range of ca. 920-1050 °C. Nevertheless, the calculated temperatures are the lowest in the fine-grained rocks regardless of the calculation method.

## 6 Discussion

### 6.1 The role of strain localization and melt infiltration

The Oran mantle xenoliths show a variable grain size ranging from coarse granular to fine-grained equigranular textures uncorrelated with modal variations (Fig. 2a-b). In polymineralic mantle rocks, second phases such as pyroxenes or spinel, are well known to keep steady state grain size of dynamically recrystallized olivine small by pinning or dragging the grain boundaries of the matrix phase (e.g., Warren and Hirth, 2006;Precigout et al., 2007;Skemer et al., 2010;Linckens et al., 2011b). In fact, the olivine grain size in the Oran peridotites show systematic correlations with the ratio between pyroxene grain size

and volume fraction (i.e., Zener parameter; see Herwegh et al., 2011 for a review) (Fig. 8a-b). Since the fraction of secondary phases other than pyroxenes is very low (<1%; Table 1) in the studied xenoliths, we include only olivine and pyroxenes in our analysis, but we plot ortho- and clinopyroxene separately to distinguish potential differences in their effect on olivine grain size. Dynamic recrystallization —which is a common grain size reduction process in dislocation accommodated creep (e.g., De Bresser et al., 2001)—, is apparent in the microstructure of the coarse-grained xenoliths if orthopyroxene is considered as

a second phase particle (Fig. 8a). In contrast, in case of clinopyroxene being a second phase particle, the coarse-grained peridotites plot along the transition between dynamic recrystallization-controlled and second phase-controlled fields (Fig. 8b).



These observations suggest that in the coarse-grained peridotites olivine yielded a relatively stable grain size by dynamic recrystallization, which was moderately influenced by the presence of clinopyroxene and it was not pinned by orthopyroxene. In the fine-grained microstructures olivine grain size strongly depends on the Zener parameter (Fig. 8a-b), implying that small interstitial ortho- and clinopyroxenes dispersed in the fabric impeded olivine grain growth through pinning. This data is

consistent with that in previous studies, where olivine grain size was interpreted as controlled by pyroxenes through pinning during ductile deformation and strain localization processes (e.g., Warren and Hirth, 2006;Herwegh et al., 2011;Linckens et al., 2011a;Linckens et al., 2011b;Tasaka and Hiraga, 2013;Tasaka et al., 2014;Hansen and Warren, 2015).

Grain size reduction in mantle xenoliths is symptomatic of increasing strain localization in a relatively cold and then stiffer and high stress domains in the mantle lithosphere (e.g., Drury et al., 1991;Karato, 2008;Linckens et al., 2011b;Vauchez et al.,

2012;Tommasi and Vauchez, 2015). Strain localization in peridotites is often linked to mantle metasomatism because fine-grained microstructures frequently display relative enrichment of incompatible trace elements (e.g., Yang et al., 2010) or evidence of modal metasomatism (e.g., Newman et al., 1999;Dijkstra et al., 2002). The relative timing between strain localization and metasomatism is however not always evident. Detailed studies of the textures and CPO of olivine and pyroxenes helps to unravel if the texture is consistent with solid-state or melt-present deformation (Tommasi et al., 2008;Higgie

and Tommasi, 2014) that are of interest to decipher the origin and significance of grain size variations in the Oran xenolith suite.

### 6.1.1 Coarse-grained peridotites

In Oran coarse-grained peridotites, olivine and orthopyroxene show CPO coherent with coeval deformation by dislocation creep at high temperature. The alignment of $[100]_{Ol}$ and $[001]_{Opx}$ crystallographic axes (Fig. 5a) subparallel to the stretching

lineation is consistent with dominant activation of (0kl)[100] slip systems in olivine and the (100)[001] slip system in orthopyroxene. Rotation axes accommodating low-angle (2–15°) misorientations of <0vw> orientations in olivine and around [010] in orthopyroxene (Fig. 6) corroborate the activation of these slip systems (de Kloe, 2001). These slip systems are dominant in crystalplastic deformation under low stress, high temperature (>1000 °C), low pressure and fluid-absent conditions (cf., Carter and Avé Lallemant, 1970;Durham and Goetze, 1977;Mainprice et al., 2005;Karato, 2008;Jung et al.,

2010;Demouchy et al., 2014). The fine-grained pyroxenes in the matrix are strain-free and elongated (Fig. 3c and close-up view), and they occur in interstitial textural position at olivine-olivine grain boundaries (Fig. 3b). These features are inconsistent with neoblast produced during dynamic recrystallization of porphyroclast (Urai et al., 1986). Similar fine-grained pyroxenes in coarse-grained peridotites are interpreted as the result of post-kinematic crystallization of secondary pyroxene from small melt fraction (Tommasi et al., 2008;Morales and Tommasi, 2011). The similar CPO and rotation axes of small and

large orthopyroxene grains in Oran coarse-grained xenoliths (Fig. 5-6) indicate that interstitial orthopyroxenes deformed coherently with olivine and orthopyroxene porphyroclasts by dislocation creep. This observation excludes a post-kinematic crystallization origin for the fine-grained matrix pyroxenes. Their textural position (Fig. 3b) are consistent with crystallization after small-fraction of interstitial melts between olivine–olivine boundaries. The texture and CPO of fine-grained pyroxenes

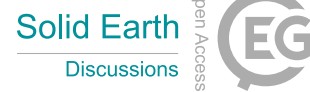

can better be accounted for if, upon crystallization from small melt fractions along olivine grain boundaries, they deformed by dislocation creep, followed by a last post-kinematic recovery stage. The Oran coarse-grained texture is then consistent with high-T deformation at near solidus conditions and infiltration of small melt fractions. Experimental studies of Holtzman et al. (2003) show that small melt fractions in the presence of second phase particles do not result in a switch of olivine CPO up to
6% melt fraction, which is consistent with low melt fractions inferred for the formation of fine-grained pyroxenes in the coarse-grained rocks.

### 6.1.2 Fine-grained peridotites

Relative to other textural types, the equigranular peridotites display a shift of olivine CPO symmetry towards [010]-fiber, have a lower fabric strength, and the orthopyroxene CPO are inconsistent with coeval deformation by dislocation creep (Fig. 4-5;
Table 1). The fine grain size of olivine and pyroxenes is consistent with high-stress deformation likely in mantle shear zones. Several processes may result in [010]-fiber CPOs in olivine: axial compression or 3–D transpressive deformation controlled by the finite strain geometry (Nicolas et al., 1973;Tommasi et al., 1999;Chatzaras et al., 2016); static recrystallization (Tommasi et al., 2008); activation of [100] and [001] glide directions at high pressure or stress (Tommasi et al., 2000;Mainprice et al., 2005); and melt-present deformation (Holtzman et al., 2003;Le Roux et al., 2008;Higgie and Tommasi, 2012). The
preponderance of [010]-fiber like CPO in olivines of equigranular peridotite xenoliths and their fine grain size may be compatible with deformation in transpression in mantle shear zones. Transpressional deformation alone, however, hardly explains the increasing dispersion and weakening of the olivine CPO with increasing pyroxene content. The development of [010]-fiber CPO during annealing requires selective grain growth that results in tabular crystal morphologies flattened parallel to the (010) plane (Tommasi et al., 2008). This model is at odds with the overall fine grain size and polygonal olivine crystal
shapes in the equigranular textures (Fig. 3d-f). The spinel to plagioclase facies origin of the Oran xenoliths indicates that deformation took place at shallow depth, where the prevailing pressure conditions were too low for the activation of [001] glide (Mainprice et al., 2005;Vauchez et al., 2005). In small olivine grains, intracrystalline misorientations are accommodated mainly by rotations around [010] (Fig. 6a). This rotation axis is compatible with the activation of either (001)[100] or (100)[001] slip at tilt subgrain boundaries (e.g., de Kloe, 2001). The former slip system has been reported in solid-state
deformation experiments at moderate water contents (12-60 wt ppm $H_2O$), moderate stress (400 MPa) and pressure (2 GPa) (Katayama et al., 2004), but it does not disperse $[100]_{Ol}$ crystallographic axes. The latter slip may reflect high-stress deformation and can contribute to the development of [010]-fiber olivine CPO symmetry. In the lack of further microstructural evidence, we cannot differentiate between these two possibilities, but considering the higher concentration of $[100]_{Ol}$ axes with respect to $[001]_{Ol}$ in every sample, we suspect that dominant activation of [001] glide is less likely. Thus, we favor the latter
hypothesis that is shear deformation of olivine in the presence of melt to explain the shift towards [010]-fiber olivine patterns in the equigranular peridotites. In these textures, the crystallographic axes of small orthopyroxenes are weakly oriented and, occasionally, distributed subparallel to those of olivine (Fig. 5b; Fig. S1). Similar correlation between olivine and orthopyroxene CPOs in orthopyroxene-impregnated dunites from the Bay of Islands ophiolite (Suhr, 1993), in ultramylonitic



shear zones from the Othris ophiolite (Dijkstra et al., 2002) and in fluid-assisted ductile strain localization from the Ronda massif (Hidas et al., 2016b) has been interpreted as formed due to synkinematic melt/fluid-facilitated dissolution-precipitation processes. In the Oran equigranular peridotites, we interpret the CPO of interstitial small orthopyroxenes similarly, and we propose that they record constrained synkinematic crystallization from a melt phase. As in coarse-grained peridotites, the lack of intracrystalline deformation in equigranular peridotites points to a post-kinematic recovery stage, possibly enhanced by the presence of melts as proposed by Rampone et al. (2010) and Hidas et al. (2016a) in mantle xenoliths from Tallante.

The microstructure and CPO of fine-grained porphyroclastic peridotites are transitional between the coarse-grained and the equigranular peridotites (Fig. 2,4). The orthopyroxene porphyroclasts have a coherent CPO with large olivines, which can be interpreted by the same operative slip systems as those discussed earlier in the coarse-grained rocks. In the fine-grained matrix, olivine CPO is identical to that of larger olivine grains but orthopyroxene CPO is not coherent with a coeval solid-state deformation. This duality is corroborated by the analysis of intracrystalline rotation axes that show <0vw> orientations in large olivines and a predominant maximum around [010] axis in orthopyroxene porphyroclasts, similarly to the coarse-grained peridotites, and the occurrence of $[010]_{Ol}$ and $[001]_{Opx}$ axes in the small grains, similarly to the equigranular ones (Fig. 6). These observations attest for the increasing impact of melt on the deformation and, eventually, a switch of deformation mechanisms at the transition from coarse-grained to fine-grained microstructures.

Deformation in the presence of melt may result in weak olivine CPO due to partitioning of the deformation between dislocation creep and diffusion-assisted grain boundary sliding (Holtzman et al., 2003;Zimmerman and Kohlstedt, 2004;Le Roux et al., 2008). This is consistent with our microstructural observations because olivine CPO is the weakest in the equigranular peridotites that record the shift of olivine CPO symmetry towards [010]-fiber (Fig. 4). The presence of melts is further corroborated by the highest abundance of fine-grained pyroxenes in the equigranular peridotites (Table 1).

The presence of melt-related small pyroxenes in the coarse-grained rocks showing coherent CPO with olivine, the deformation mechanisms that are consistent with the presence of melts in the equigranular samples, and the transitional microstructural features of fine-grained porphyroclastic peridotites suggest synkinematic focusing of melts in the fine-grained peridotites. Such positive feedback between melt/fluid migration and strain localization is often observed in narrow ductile mantle shear zones (Downes, 1990;Kelemen and Dick, 1995;Dijkstra et al., 2002;Yang et al., 2010;Kruckenberg et al., 2013;Hidas et al., 2016b). Finally, based on the microstructural evidence for orthopyroxene replacement by clinopyroxene and the dominantly identical CPO of both pyroxenes in each xenolith, we suggest that crystallographic control by the orthopyroxene crystal structure during the crystallization of clinopyroxene resulted in a clinopyroxene CPO that follows that of orthopyroxene.

## 6.2 Record of reactive melt transport in the Oran shallow SCLM

The Oran mantle xenoliths originate from the spinel- to plagioclase facies SCLM and the major element content of their olivine and spinel indicate that they are residual peridotites produced by relatively high degrees of partial melting (10-25%; Fig. 7a). However, apart from one harzburgite, the refractory compositions (>10% of partial melting) are often coupled to lherzolite or



wehrlite lithologies (Fig. 7a). This observation suggests that the formation of clinopyroxene-rich modal compositions by melt-rock reaction proceeded in a variously depleted refractory —likely harzburgite— protolith. Modal variations (Fig. 2a) and textural evidence (Fig. 3b) in most peridotites are consistent with the hypothesis that clinopyroxene crystallization was accompanied by dissolution of orthopyroxene, and possibly minor precipitation of olivine. Based on the modal content (Table

1) and the major element geochemistry of the mineral cores (Table S1), we estimated the whole rock major element composition of the studied rocks (Table 2). Fig. 9 shows the covariation of bulk rock Mg# and Cpx/Opx ratios. In this figure we also illustrate the trend of batch partial melting of a fertile lherzolite at 0.5-2.0 GPa (adapted from Varas-Reus et al., 2016 and references therein) together with the predicted curves from numerical modeling involving precipitation of clinopyroxene ±olivine at the expense of orthopyroxene in the spinel stability field (adapted from the model of Ionov et al., 2005). Ionov et

al. (2005) modelled the interaction of refractory harzburgite (Mg#=91.0; Cpx/Opx=0.32) with percolating basaltic liquids having either a primitive, high-Mg# (0.76) or an evolved, low-Mg# (0.63) composition, involving clinopyroxene-forming reactions at decreasing melt mass.

   Apart from a few coarse-grained peridotites, batch melting in the spinel facies is inconsistent with the composition of most Oran xenoliths. To explain the composition of these rocks by batch melting processes, significantly higher pressures are

required (>3 GPa; e.g., Varas-Reus et al., 2016 and references therein), which is not compatible with the shallow SCLM origin of the samples. In contrast, the modal variations and the compositional variability of the studied xenoliths can be plausibly explained by wehrlitization. Unlike the expected high CaO/Al$_2$O$_3$ and Na$_2$O/Al$_2$O$_3$ values of wehrlites related to carbonatites (Yaxley et al., 1991;Raffone et al., 2009), the Oran xenoliths have geochemical composition (Table S1) typical of mantle wall rocks that interacted with melts relatively close to magma conduits or porous flow channels (Bodinier et al., 1990;Ionov et al.,

2005). The compositions of most coarse-grained mantle peridotites are consistent with interaction with a high-Mg# liquid at increasing melt-rock ratios from lherzolite towards wehrlite (Fig. 9). Similarly, the composition of fine-grained lherzolites can be explained by interaction with a low-Mg# melt at low melt-rock ratio (Fig. 9). Comparison with numerical modelling suggests that the low Mg# of a fine-grained lherzolite (sample HAM-018) may reflect interaction with a low-Mg# melt at high melt/rock ratio. The composition of the coarse-grained wehrlite from Souahlia plots in between the results of the two numerical

modelling provided in Ionov et al. (2005) (Fig. 9). This suggests either an interaction with a liquid having Mg# intermediate between the modelled cases, or interaction with a low-Mg# melt at an increased mass ratio of crystallized minerals to infiltrated melt (R value >0.3). Even after considering the uncertainties in the orthopyroxene content in this sample —that may influence the Cpx/Opx ratios— in both cases high melt-rock ratios are required to explain the whole-rock composition of this wehrlite sample.

In summary, except for the wehrlites, the composition of coarse-grained Oran mantle peridotites suggests interaction with a high-Mg# melt at low melt/rock ratios, which is also supported by microstructural interpretation based on which deformation was accommodated under low-melt conditions in these textures. The formation of coarse-grained wehrlites requires an increase in the melt/rock ratio and an interaction with either a high-Mg# or an intermediate-Mg# melt. In contrast, the geochemical composition of many fine-grained Oran xenoliths, mostly lherzolites, is consistent with interaction with low-Mg# melts. These





observations point to a change in the melt composition during melt-rock interaction. Primitive melts can evolve towards lower Mg# upon crystallization and cooling (e.g., Pichavant and Macdonald, 2007 and references therein), thus the composition of the coarse-grained Oran peridotites suggests interaction with a less fractionated melt at depth, likely at higher temperatures, while the composition of the equigranular peridotites can be addressed by interaction with a highly fractionated, evolved liquid

of the primitive melt at shallower depths at increasing melt/rock ratios. This interpretation is supported by the slightly lower calculated temperature of the equigranular xenoliths compared to the coarse-grained ones (Fig. 7b).

## 6.3 Role of the STEP fault in melt-rock interaction in the shallow SCLM

In the Cenozoic geodynamic evolution of the western Mediterranean, thinning of the SCLM and crust, as well as opening of a slab window beneath the N-African margin occurred during the late Miocene rapid westward retreat of the subduction system

along the Rif-Tell STEP fault (e.g., Duggen et al., 2005;Chertova et al., 2014a;Chertova et al., 2014b and references therein) (Fig. 10). The slab rollback and continental edge delamination resulted in a strong subduction suction causing the Canary plume material to flow ≥1500 km to the northeast (Duggen et al., 2009). This lateral mantle flow is supported by the seismic anisotropy signal beneath NW Africa that suggests E-W to NE-SW directed mantle flow (Diaz et al., 2010), as well as by the geochronological data and geochemical imprint of the Si-poor intraplate magmatism that shows northeastward-younging ages

in NW Algeria (from ca. 7 Ma to 0.8 Ma; Duggen et al., 2005). The position of the Oran volcanic field in the western Mediterranean suggests that its SCLM records processes related to mantle upwelling, lateral mantle flow, lithospheric tearing and formation of STEP fault associated with the westward slab rollback of the subduction front (Fig. 10).

Upwelling of hot mantle material coupled to extension potentially eroded the base of the SCLM and resulted in the formation of large amounts of $SiO_2$-undersaturated, high-Mg# alkaline melts at the N-African margin (Duggen et al.,

2005;Duggen et al., 2009;Varas-Reus et al., 2017). During influx of such melts in refractory peridotites at mantle conditions (>1000 °C and 1.0 GPa), the first reaction to take place is the dissolution of orthopyroxene and simultaneous crystallization of clinopyroxene and olivine (Shaw, 1999;Shaw et al., 2005;Lambart et al., 2012;Marchesi et al., 2017). Close to melt percolation channels, the interaction between the high-Mg# melts and the harzburgite protolith proceeds until the complete exhaustion of orthopyroxene in the rocks, forming coarse-grained wehrlites (e.g., SOU-003 wehrlite; Fig. S1) that record high melt/rock

ratios in the Oran xenoliths (Fig. 9). The residual melt of the reaction has a higher alkaline, $SiO_2$, $Al_2O_3$- and CaO-rich content with respect to the primitive high-Mg# melt. Rapid Fe-Mg exchange between mantle olivine and the melt (Gaetani and Watson, 2002;Shaw, 2004) might partially buffer the MgO/FeO ratio in the residual melt and resulted in a decreasing Mg# of olivine in mantle peridotites higher in the SCLM column. This stage is reflected in the low Mg# (<90.0) of some wehrlite and lherzolite xenoliths (Fig. 9). The evolution of the residual melt also leads the system towards saturation in orthopyroxene at shallower

SCLM conditions. At the shallowest levels of the SCLM represented by the equigranular xenoliths, the highly evolved melts became saturated in orthopyroxene and the interaction resulted in the precipitation of small interstitial orthopyroxenes. Considering that small pyroxenes exert a strong influence on olivine grain size in the fine-grained mantle xenoliths (Fig. 8a-b), and the shift of olivine CPO towards [010]-fiber symmetry in some equigranular xenoliths suggests the presence of melts



during deformation, we propose that this reaction stage was synkinematic. High time-integrated melt/rock ratios are plausibly accomplished by melt focusing (Phipps Morgan, 1987;Asimow and Stolper, 1999), which is compatible with synkinematic melt-rock interaction in a ductile shear zone. We therefore propose that coarse-grained xenoliths record deformation at moderate to high temperature by dislocation creep in a relatively dry or at least low-melt system, at the base of the studied

SCLM column and farther from melt channels. This deformation may either be an older event preserved in the rocks, or correspond to the lateral mantle flow from the Canary plume. As recorded in the porphyroclastic, and particularly in the fine-grained equigranular xenoliths, at shallower levels of the SCLM, deformation was accommodated in the presence of increasing amounts of melts. This deformation is probably related to the westward propagation of the western Mediterranean subduction system along the Rif-Tell STEP fault parallel to the N-African margin in the Miocene (Duggen et al., 2005;Chertova et al.,

2014a;Chertova et al., 2014b), which resulted in the formation of ductile shear zones in the shallow SCLM (Fig. 10). These results suggest that wehrlitization of the Oran SCLM is a relatively young event, which took place in late Miocene to early Pleistocene times.

## 7 Conclusions

The modal variation, microstructure and major element composition of mantle xenoliths from the Oran volcanic field (NW

Algeria) are consistent with synkinematic melt-rock interaction in the shallow subcontinental lithospheric mantle. Coarse-grained mantle xenoliths record coherent olivine and orthopyroxene CPO that developed at high temperature by dislocation creep in a dry or low-melt system. In the fine-grained microstructures, olivine CPO records a shift towards [010]-fiber symmetry indicating the presence of melts during deformation, and orthopyroxene CPO is not compatible with solid-state deformation. Clinopyroxene inherits the orientation of orthopyroxene during orthopyroxene-consuming and clinopyroxene-

precipitating melt-rock reaction. The strong exert of small pyroxene crystals on the olivine grain size suggests their presence during deformation and points to channelling of reactive melts at shallow subcontinental lithospheric mantle beneath the N-African margin. The textural and geochemical record of the peridotites are consistent with interaction of a refractory harzburgite protolith with a high-Mg# melt at depth (resulting in the formation of coarse-grained clinopyroxene-rich lherzolite and wehrlite), and with a low-Mg# evolved melt in the shallow subcontinental lithospheric mantle (forming fine-grained

harzburgite). Based on the microstructural evidence the melt-rock interaction was likely synkinematic. We propose that reactive melts formed during the operation of the Rif-Tell subduction-transform edge propagator (STEP) fault beneath the N-African margin during the Miocene westward retreat of the western Mediterranean subduction system.





## 8 Acknowledgements

We acknowledge the help of Z. Konc in the EPMA and EBSD data acquisition. We are grateful to D. Mainprice (Géosciences Montpellier, France) for providing scripts and help in MTEX, and to L.E. Aradi (Lithosphere Fluid Research Lab, Eötvös Loránd University of Budapest, Hungary) for contributing to an earlier version of the EBSD database.

Research leading to these results was funded by the "Agencia Estatal de Investigación" (AEI, Spain) grants CGL2016-75224-R, CGL2016-81085-R and CGL2015-67130-C2-1-R, by the "Junta de Andalucía" research groups RNM-131 and RNM-148 and the International Lithosphere Program (CC4-MEDYNA). K.H. acknowledges funding by "Ministerio de Economía y Competitividad" (MINECO) (FPDI-2013-16253) and the AEI (CGL2016-81085-R). C.M. acknowledges funding by Ramón y Cajal Fellowship RYC-2012-11314 by the MINECO.

Fellowships, research and infrastructure grants leading to this research have been co-funded by the European Social Fund (ESF) and the European Regional Development Fund (ERFD) of the European Commission.

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



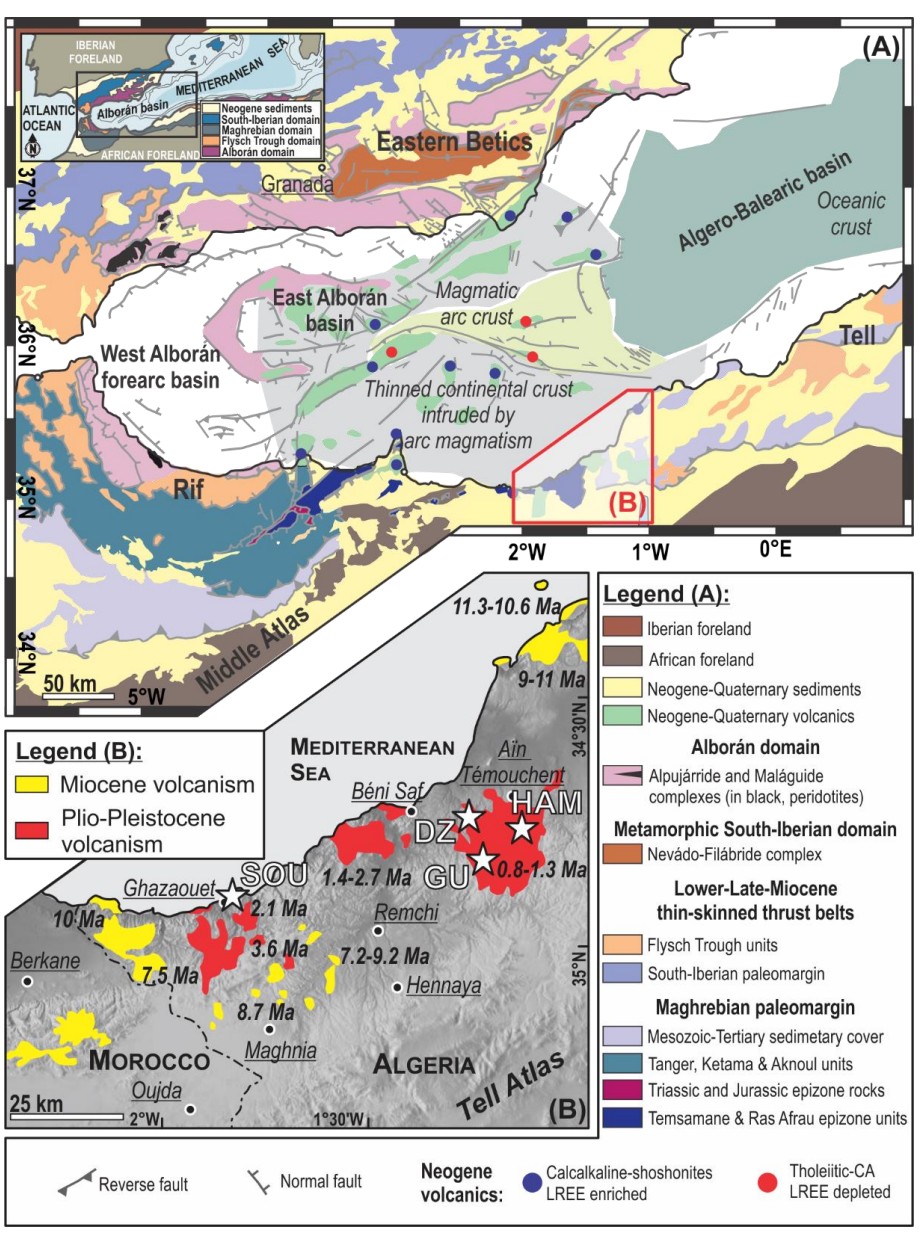

**Figure 1: (a) Simplified geological map of the westernmost Mediterranean, showing the main tectonic domains of the Betics-Rif orogenic belt with the areas of oceanic crust, magmatic arc and thin continental crust in the Alborán basin (modified after Booth-Rea et al., 2007 and references therein). Area in a red frame indicate the Oran volcanic field, and it is enlarged in (b). (b) Schematic map of the Oran volcanic field showing the occurrences of Miocene and Plio-Pliocene volcanic rocks. Italic numbers correspond to the age of volcanism after Maury et al. (2000) and Duggen et al. (2005). White stars indicate the location of the studied mantle xenoliths with the prefix of the sample labels as DZ (Djebel Dzioua), GU (Djebel Gueriane), HAM (Hammar E'Zohra) and SOU (Souahlia).**





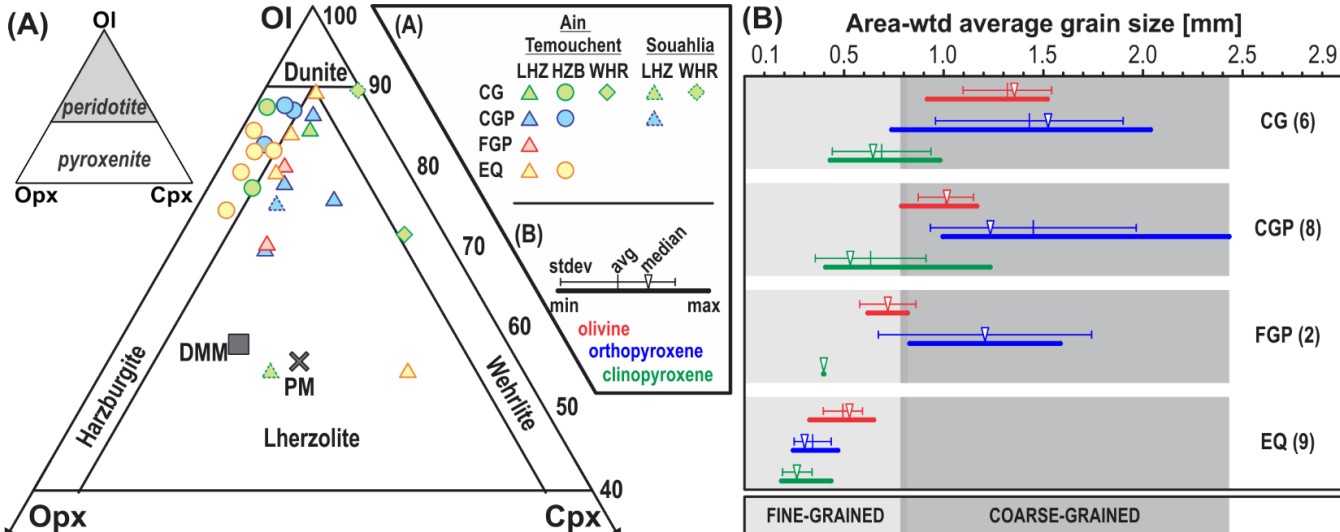

**Figure 2: (a) The modal composition of the Oran mantle xenoliths in a Streckeisen diagram for ultramafic rocks (Streckeisen, 1976). DMM: depleted MORB mantle and PM: primitive mantle (after McDonough and Sun, 1995; Workman and Hart, 2005). (b) Distribution of area-weighted mean grain size of olivine, orthopyroxene and clinopyroxene in the Oran mantle xenoliths. Horizontal thick bars are the full range of grain sizes (from minimum to maximum) in a given textural group, thinner horizontal bars show the average with standard deviation and white triangles indicate the median value. Numbers in brackets give the number of xenoliths in a given textural group. CG: coarse granular; CGP: coarse-grained porphyroclastic; FGP: fine-grained porphyroclastic; EQ: fine-grained equigranular. See text for further details.**



**Figure 3: Representative textures of a coarse-grained (a-c) and a fine-grained (d-f) mantle xenolith from Oran. In each row the same area is shown in cross-polarized optical images (a,d), EBSD phase maps (b,e) and misorientation to the mean orientation of the grain (Mis2Mean) maps (c,f; units are in degrees). White triangle in (a-c) points at a subgrain boundary in olivine. Inset in (b) shows the crystallographic orientation of an orthopyroxene crystal rimmed by clinopyroxene in lower hemisphere equal-area projection of the main crystallographic axes (n denotes the number of orientations plotted for each crystal).**





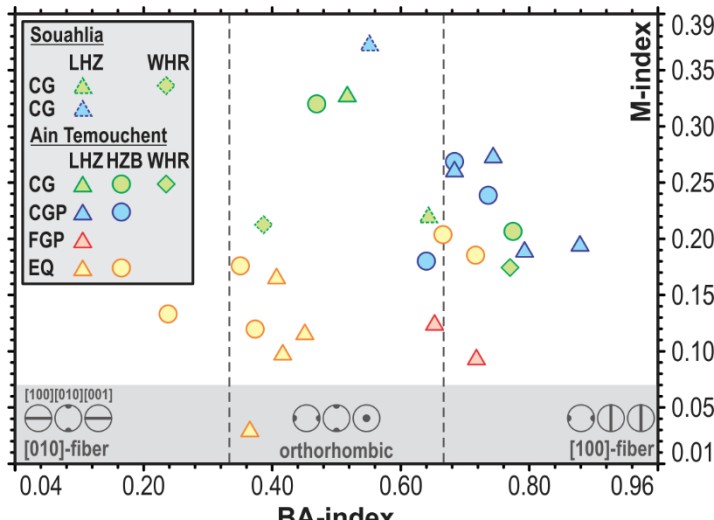

**Figure 4: BA-index of olivine in the function of olivine fabric strength (expressed as M-index). Vertical dashed lines show the guidelines for differentiating [010]-fiber, orthorhombic and [100]-fiber CPO symmetries. Abbreviations in the legend are the same as for Table 1.**



**Figure 5: Crystal-preferred orientation (CPO) of olivine, orthopyroxene and clinopyroxene in representative coarse-grained (a) and fine-grained mantle xenoliths from the Oran volcanic field. For the full data set visit supporting information Figure S1. Pole figures are lower hemisphere, equal area stereographic plots using one average measurement per grain ("one point per grain") with contours at 0.5 multiples of a uniform distribution. In oriented thin sections of DZ-003 and HAM-005b the horizontal line denotes the foliation, and a white star corresponds to the lineation. The CPO in other xenoliths has been rotated in this reference frame in order to make the comparison possible. See text for further details on this rotation. N, number of measured grains; pfJ, scalar measure of the strength of the axis orientation in the pole figure. To the left of the pole figures, the characteristic microstructure is shown for each sample as EBSD phase map (color-coding is the same as in Fig. 3) with the same magnification.**



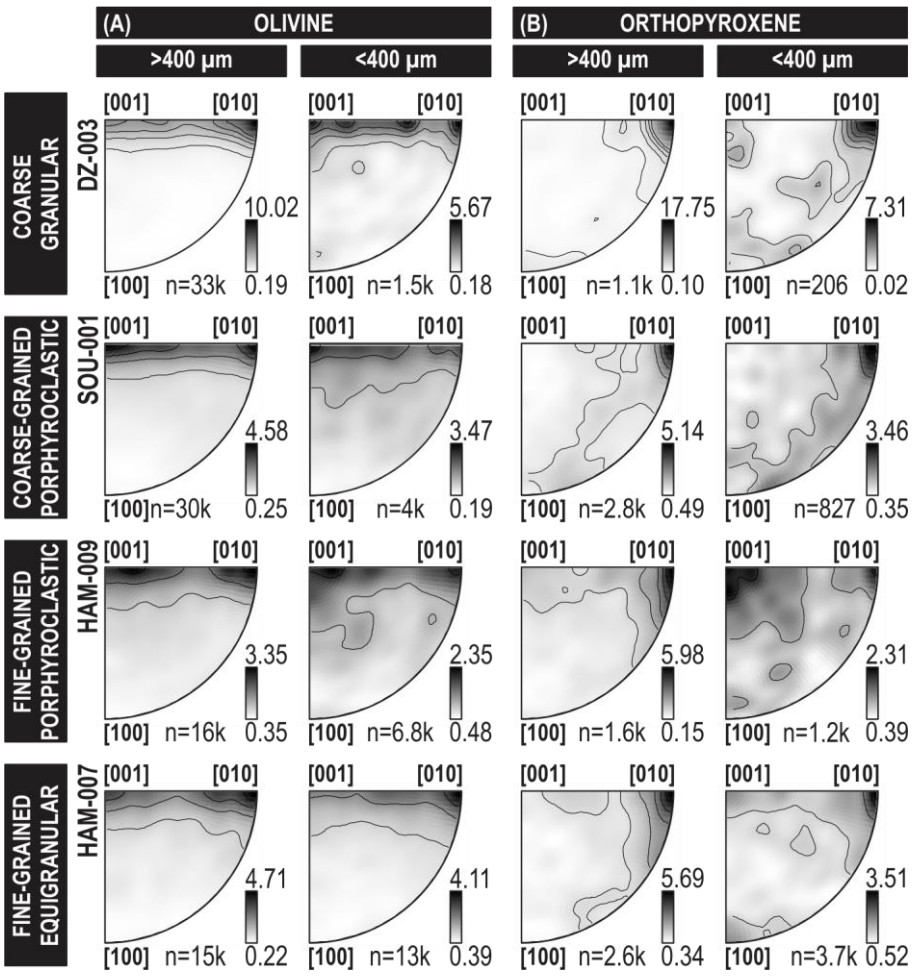

**Figure 6: Rotation axes accommodating low-angle (2-15°) misorientations in the large (>400 µm equivalent diameter) and small (<400 µm equivalent diameter) grain fraction of olivine (a) and orthopyroxene (b). Data is represented in inverse pole figures with contouring at 0.5 multiples of uniform distribution. N denotes the number of pixels plotted.**



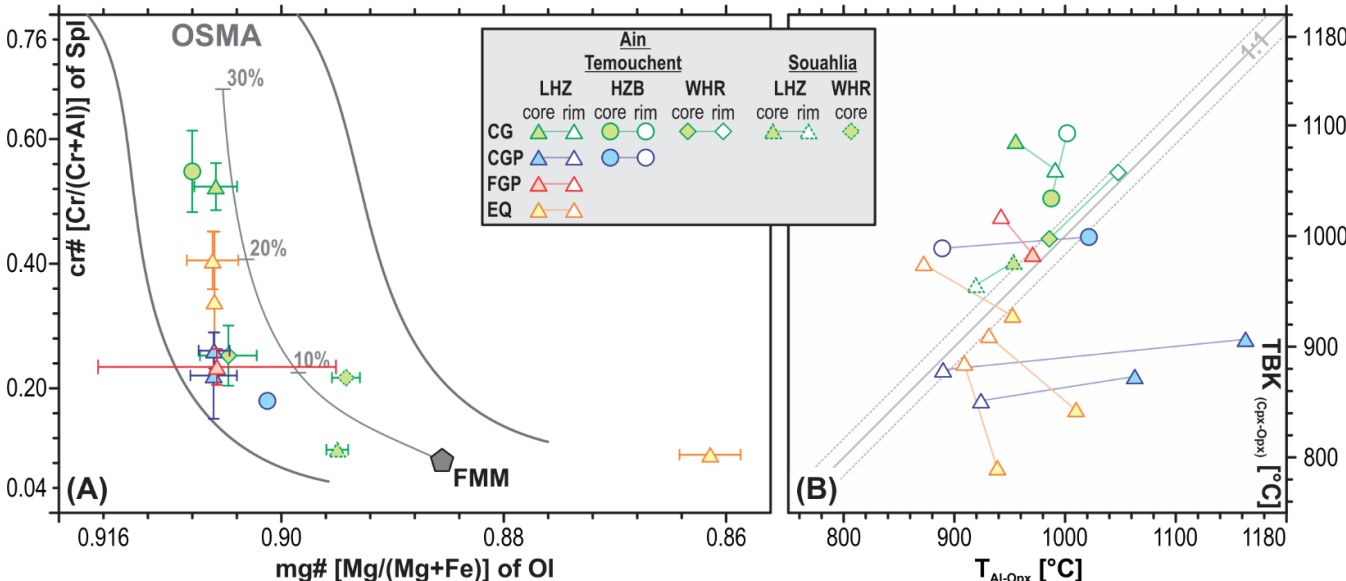

**Figure 7: (a) Plot of spinel Cr# [Cr/(Cr+Al)] versus olivine Mg# [Mg/(Mg+Fe)] expressed as atom per formula units. The Olivine-Spinel Mantle Array (OSMA) and degrees of partial melting are from Arai (1994). FMM (grey pentagon) is the Fertile MORB Mantle (after Pearce et al., 2000). Ol: olivine; Spl: spinel. (b) Estimated temperatures of Oran mantle peridotites based on the Cr-Al-orthopyroxene (T$_{Al-Opx}$) geothermometer of Witt-Eickschen and Seck (1991), and the two-pyroxene solvus (T$_{Cpx-Opx}$) geothermometer of Brey and Köhler (1990). Data presented in Table 1; see abbreviations there.**



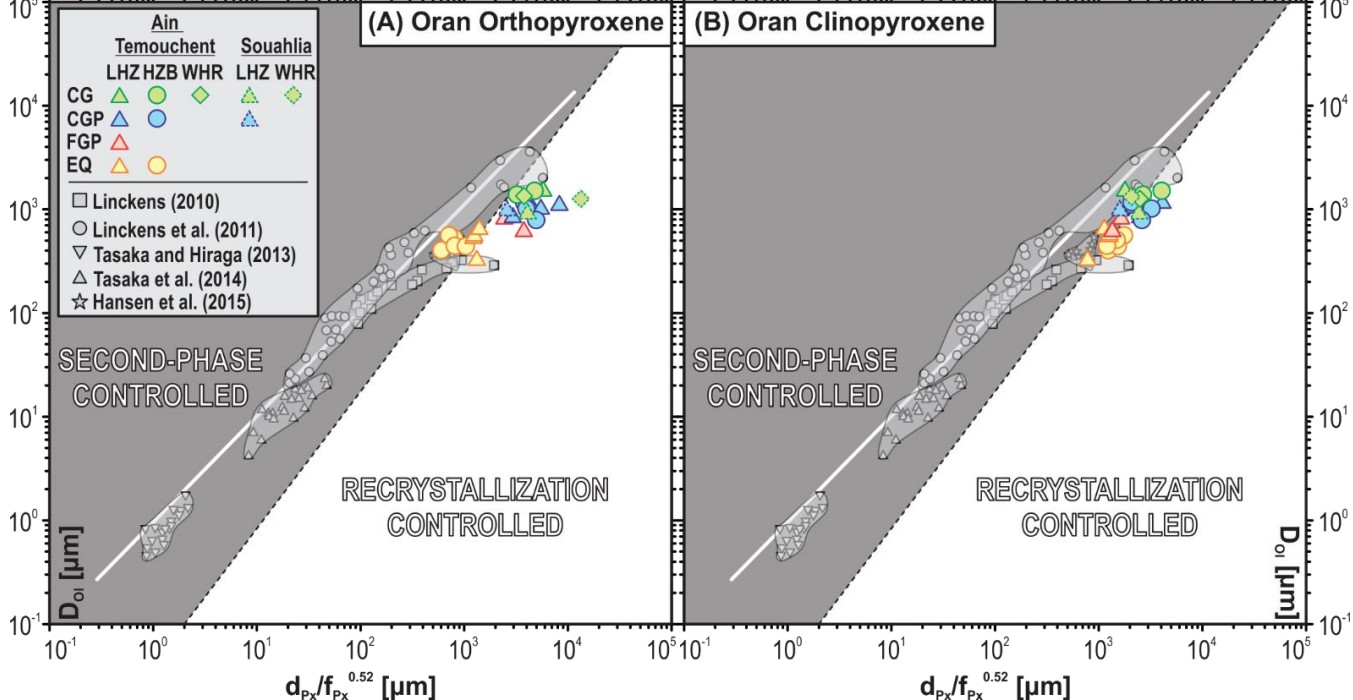

**Figure 8: (a) Zener diagram showing average area-weighted olivine grain size ($D_{Ol}$) as a function of grain size and volume fraction of pyroxene ($d_{Px}/f_{Px}^{0.52}$) in the Oran mantle xenoliths. Data is plotted considering orthopyroxene (a) and clinopyroxene (b) as secondary phases in the olivine matrix. The diagram shows the transition from second-phase (grey-shaded field) to dynamic recrystallization (white-shaded field) controlled microstructures after Herwegh et al. (2011). Comparative data is from natural peridotite shear zones in the Oman ophiolite (Linckens, 2010;Linckens et al., 2011a;Linckens et al., 2011b;Tasaka et al., 2014) and the Josephine Peridotite (Hansen et al., 2015), as well as deformation experiments on olivine-pyroxene aggregates (Tasaka and Hiraga, 2013). For plotting the data a Zener parameter of Z=0.52 is used following Tasaka et al. (2014) and the solid white line in the background is the fit presented in their work.**



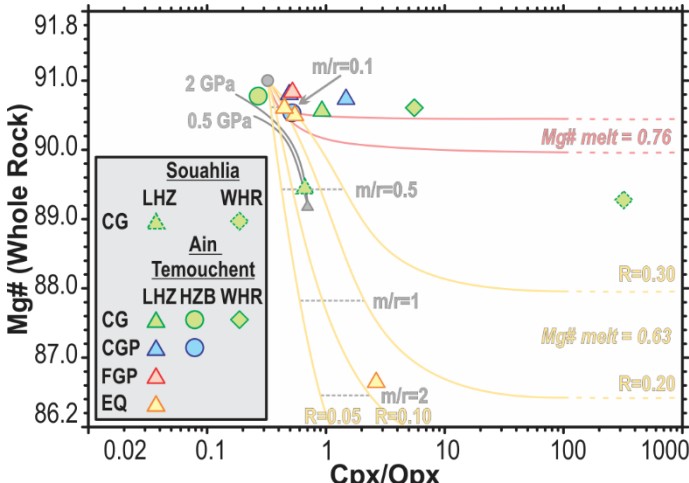

**Figure 9: Clinopyroxene/orthopyroxene (Cpx/Opx) ratio of representative Oran mantle peridotites in the function of their calculated whole rock Mg#. Data are compared to the results of modelling of batch partial melting in the range of 0.5-2.0 GPa (Varas-Reus et al., 2016 and references therein) and the variations produced by interaction of a refractory peridotite (Mg#=0.91; Cpx/Opx=0.32) with percolating high-Mg# (0.76) and low-Mg# (0.63) basaltic liquid involving Cpx-forming reactions at decreasing melt mass (Ionov et al., 2005). For the low-Mg# melt, R-values indicate the mass ratio of crystallized minerals to infiltrated melt ranging from 0.02 to 0.3, and gray dashed lines denote the melt-rock (m/r) ratio. For the high-Mg# melt composition, all modelling results plot between the red lines. Dashed lines indicate extrapolation from original numerical modeling presented in Ionov et al. (2005). See text for further details and Table 1 for abbreviations shown in the legend.**



**Figure 10: (a) Conceptual 3-D model of the western Mediterranean lithosphere, stripped from the overlying crust (modified after Mancilla et al., 2015). The original sketch showed the geometry of the lithosphere and its relation with the subducted oceanic slab as observed by tomographic studies (Spakman and Wortel, 2004; García-Castellanos and Villaseñor, 2011; Bezada et al., 2013). MA: Málaga; GR: Granada; ND: Nador. The superposed schematic geological map is presented in Fig. 1a. Types and ages of volcanism are compiled after Maury et al. (2000), Duggen et al. (2005) and Booth-Rea et al. (2007). On the left, seismic tomography is shown for an E-W cross section close to the Oran volcanic field (light blue dashed line in the 3-D sketch) after van Hinsbergen et al. (2014). Brown, orange and yellow solid lines indicate the estimated position of the subduction front at 13 Ma, 7 Ma and 0 Ma, respectively (Booth-Rea et al., 2007 and pers. comm. 2018). The Betic and Rif-Tell STEP faults are constrained after Hidas et al. (2016a), Mancilla et al. (2018), and Chertova et al. (2014a,b). Mantle flow in the Iberian and N-African slab windows is inspired by the work of Menant et al. (2016). White star shows the provenance of Oran mantle xenoliths.**



| Sample | Lithology (Fabric)[a] | Estimated Temperature [°C][b] | | | | | | Modal Composition [%] | | | | | | Area-wtd Mean Grain Size [mm] | | | Calculated Parameters[c] | | | |
| | | core | | | rim | | | | | | | | | | | | | | | |
| | | $T_{Ca-Opx}$ | $T_{Al-Opx}$ | $T_{solv}$ | $T_{Ca-Opx}$ | $T_{Al-Opx}$ | $T_{solv}$ | Ol | Opx | Cpx | Spl | Amp | Plag | Ol | Opx | Cpx | BA | $M_{Ol}$ | $M_{Opx}$ | $M_{Cpx}$ |
|---|---|---|---|---|---|---|---|---|---|---|---|---|---|---|---|---|---|---|---|---|
| DZ-002 | HZB(CG) | 950 | 1021 | 1034 | 1028 | 1002 | 1093 | 77.0 | 17.7 | 4.8 | 0.6 | | | 1.4 | 1.3 | 0.5 | 0.47 | 0.32 | 0.27 | 0.05 |
| DZ-003 | LHZ(CG) | 1087 | 955 | 1086 | 1108 | 991 | 1060 | 84.4 | 7.9 | 7.3 | 0.5 | | | 1.5 | 1.5 | 0.5 | 0.52 | 0.33 | 0.31 | 0.12 |
| DZ-004 | HZB(CG) | | | | | | | 87.3 | 11.1 | 1.3 | 0.2 | | | 1.5 | 1.5 | 0.4 | 0.77 | 0.21 | 0.24 | 0.22 |
| DZ-005 | LHZ(CGP) | | | | | | | 86.1 | 6.6 | 6.7 | 0.7 | | | 1.0 | 1.3 | 0.5 | 0.74 | 0.27 | 0.36 | 0.06 |
| DZ-007 | HZB(CGP) | 1023 | 987 | 999 | 967 | 889 | 989 | 86.6 | 8.5 | 4.4 | 0.6 | | | 1.2 | 1.0 | 0.4 | 0.74 | 0.24 | 0.25 | 0.03 |
| DZ-009 | LHZ(CGP) | 957 | 1063 | 873 | 911 | 924 | 851 | 69.5 | 20.0 | 9.9 | 0.6 | | | 1.2 | 1.8 | 1.2 | 0.79 | 0.19 | 0.15 | 0.14 |
| DZ-010b | LHZ(CGP) | | | | | | | 77.1 | 13.8 | 7.8 | 1.0 | 0.4 | | 0.8 | 1.1 | 0.7 | 0.88 | 0.20 | 0.21 | 0.05 |
| GU-001 | LHZ(CGP) | 1047 | 1163 | 907 | 901 | 890 | 879 | 75.1 | 9.5 | 14.0 | 1.2 | 0.3 | | 1.1 | 2.4 | 0.8 | 0.68 | 0.26 | 0.32 | 0.08 |
| GU-002 | LHZ(EQ) | | | | | | | 89.7 | 4.0 | 6.2 | 0.0 | 0.1 | 0.1 | 0.6 | 0.3 | 0.3 | 0.41 | 0.17 | 0.04 | 0.03 |
| GU-003 | HZB(CGP) | | | | | | | 82.8 | 13.7 | 3.4 | 0.0 | | | 0.8 | 1.8 | 0.4 | 0.68 | 0.27 | 0.29 | 0.09 |
| GU-007 | HZB(CGP) | | | | | | | 87.4 | 9.1 | 3.1 | 0.3 | | | 1.0 | 1.1 | 0.5 | 0.64 | 0.18 | 0.17 | 0.20 |
| HAM-001 | WHR(CG) | 1043 | 986 | 997 | 1068 | 1048 | 1057 | 71.4 | 4.3 | 23.9 | 0.4 | | | 1.3 | 0.7 | 1.0 | 0.77 | 0.17 | 0.16 | 0.07 |
| HAM-004 | HZB(EQ) | | | | | | | 78.3 | 17.8 | 2.6 | 1.2 | 0.1 | 0.1 | 0.4 | 0.2 | 0.2 | 0.72 | 0.19 | 0.06 | 0.02 |
| HAM-005b | LHZ(FGP) | 949 | 971 | 983 | 939 | 942 | 1017 | 80.1 | 12.5 | 6.6 | 0.6 | 0.0 | 0.2 | 0.8 | 0.8 | 0.4 | 0.65 | 0.12 | 0.03 | 0.03 |
| HAM-006 | HZB(EQ) | | | | | | | 81.7 | 15.3 | 2.8 | 0.2 | | | 0.5 | 0.3 | 0.2 | 0.37 | 0.12 | 0.03 | 0.05 |
| HAM-007 | HZB(EQ) | | | | | | | 74.0 | 21.8 | 3.4 | 0.8 | | 0.1 | 0.4 | 0.5 | 0.3 | 0.24 | 0.13 | 0.01 | 0.03 |
| HAM-009 | LHZ(FGP) | | | | | | | 69.8 | 19.0 | 9.4 | 1.0 | | 0.8 | 0.6 | 1.6 | 0.4 | 0.72 | 0.09 | 0.08 | 0.02 |
| HAM-011 | HZB(EQ) | | | | | | | 80.8 | 13.0 | 4.7 | 0.5 | 0.6 | 0.4 | 0.4 | 0.3 | 0.2 | 0.35 | 0.18 | 0.01 | 0.02 |
| HAM-012 | LHZ(EQ) | 939 | 939 | 790 | 979 | 909 | 884 | 83.5 | 10.1 | 5.5 | 0.6 | 0.2 | 0.1 | 0.5 | 0.4 | 0.3 | 0.45 | 0.12 | 0.03 | 0.02 |
| HAM-016 | HZB(EQ) | | | | | | | 81.9 | 13.6 | 1.4 | 3.0 | 0.1 | | 0.6 | 0.3 | 0.2 | 0.67 | 0.20 | 0.05 | 0.04 |
| HAM-017 | LHZ(EQ) | 945 | 953 | 928 | 992 | 872 | 974 | 79.0 | 14.1 | 6.3 | 0.6 | | | 0.6 | 0.5 | 0.3 | 0.42 | 0.10 | 0.02 | 0.02 |
| HAM-018 | LHZ(EQ) | 985 | 1010 | 842 | 974 | 931 | 909 | 54.1 | 12.2 | 32.4 | 0.0 | 0.2 | 0.2 | 0.3 | 0.4 | 0.4 | 0.37 | 0.03 | 0.01 | 0.01 |
| SOU-001 | LHZ(CGP) | | | | | | | 75.0 | 16.0 | 8.2 | 0.8 | | | 1.0 | 1.0 | 0.4 | 0.55 | 0.37 | 0.29 | 0.06 |
| SOU-002 | LHZ(CG) | 950 | 953 | 977 | 965 | 919 | 956 | 53.2 | 26.3 | 17.5 | 2.9 | 0.1 | | 0.9 | 2.0 | 1.0 | 0.64 | 0.22 | 0.14 | 0.04 |
| SOU-003 | WHR(CG) | - | - | | - | - | | 89.5 | 0.0 | 9.7 | 0.8 | | | 1.3 | 0.2 | 0.7 | 0.39 | 0.21 | | 0.06 |

[a]Lithology: HZB, harzburgite; LHZ, lherzolite; WHR, wehrlite/(Fabric): CG, coarse granular; CGP, coarse-grained porphyroclastic; FGP, fine-grained porphyroclastic; EQ, fine-grained equigranular. [b]$T_{Ca-Opx}$: Ca-in-orthopyroxene (±16°C), Brey and Köhler (1990); $T_{Al-Opx}$: Cr-Al-orthopyroxene (±15°C), Witt-Eickschen and Seck (1991); $T_{solv}$: clinopyroxene-orthopyroxene geothermometer (±16°C), Brey and Köhler (1990). [c]BA, BA-index of olivine; $M_{Ol}$, M-index of olivine; $M_{Opx}$, M-index of orthopyroxene; $M_{Cpx}$, M-index of clinopyroxene (Skemer et al., 2005).

**Table 1: Summary of petrography, estimated temperatures, modal composition and calculated microstructural parameters in the Oran mantle xenoliths.**





|  | DZ-002 | DZ-003 | DZ-007 | DZ-009 | GU-001 | HAM-001 | HAM-005b | HAM-012 | HAM-017 | HAM-018 | SOU-002 | SOU-003 |
|---|---|---|---|---|---|---|---|---|---|---|---|---|
| $SiO_2$ | 43.9 | 41.6 | 42.3 | 43.7 | 42.3 | 43.9 | 43.2 | 41.2 | 43.2 | 44.4 | 44.3 | 34.8 |
| $TiO_2$ | 0.02 | 0.04 | 0.01 | 0.11 | 0.07 | 0.06 | 0.02 | 0.06 | 0.08 | 0.11 | 0.17 | 0.03 |
| $Al_2O_3$ | 0.92 | 0.57 | 0.72 | 2.17 | 1.79 | 1.23 | 1.06 | 0.95 | 0.96 | 2.16 | 4.42 | 0.40 |
| $Cr_2O_3$ | 0.48 | 0.36 | 0.18 | 0.44 | 0.57 | 0.31 | 0.30 | 0.26 | 0.41 | 0.27 | 0.60 | 0.18 |
| FeO* | 7.88 | 8.70 | 8.52 | 7.47 | 8.10 | 8.24 | 8.03 | 8.52 | 8.08 | 10.3 | 7.74 | 9.57 |
| MnO | 0.14 | 0.12 | 0.14 | 0.18 | 0.15 | 0.15 | 0.15 | 0.12 | 0.11 | 0.23 | 0.14 | 0.22 |
| NiO | 0.30 | 0.35 | 0.34 | 0.32 | 0.30 | 0.28 | 0.34 | 0.34 | 0.28 | 0.20 | 0.24 | 0.32 |
| MgO | 43.5 | 46.9 | 45.7 | 41.4 | 44.5 | 44.6 | 44.7 | 45.6 | 43.7 | 37.4 | 36.8 | 44.7 |
| CaO | 3.85 | 1.79 | 1.86 | 4.30 | 2.21 | 1.21 | 2.77 | 2.36 | 3.10 | 2.96 | 5.45 | 0.08 |
| $Na_2O$ | 0.16 | 0.07 | 0.11 | 0.32 | 0.15 | 0.06 | 0.14 | 0.11 | 0.17 | 0.14 | 0.47 | 0.00 |
| $K_2O$ | 0.01 | 0.00 | 0.01 | 0.00 | 0.00 | 0.00 | 0.01 | 0.00 | 0.01 | 0.00 | 0.00 | 0.00 |
| Total | 101.2 | 100.5 | 99.9 | 100.4 | 100.2 | 100.0 | 100.7 | 99.5 | 100.1 | 98.1 | 100.3 | 90.3 |
| | | | | | | | | | | | | |
| Mg# | 0.908 | 0.906 | 0.905 | 0.908 | 0.907 | 0.906 | 0.908 | 0.905 | 0.906 | 0.867 | 0.895 | 0.893 |

**Table 2: Calculated whole rock major element composition of the Oran mantle xenoliths. FeO\*: all iron as $Fe^{2+}$; Mg#=Mg/(Mg+Fe) in mol%.**