# Peer review of "Lithosphere tearing along STEP faults and synkinematic formation of lherzolite and wehrlite in the shallow subcontinental mantle"

_Solid Earth, 2019_

## Referee Comment (RC1) · Theodoros Ntaflos (Referee) · 27 Mar 2019

The submitted to Solid Earth manuscript by Hidas et al., with title "Lithosphere trearing along STEP foults and synkinematic formation of lherzolite and wehrlite in the shallow subcontinental mantle" provides a study on mantle xenoliths demonstrating how microstructural features among their constituent minerals are related to the kind and degree of melt-rock interaction. The overall performance of this manuscript is excellent. It is well organized and the provided arguments are clear and well structured. To my knowledge it is first manuscript where throughout discussion about the relationships between melt percolation processes and deformation along a STEP fault. The authors

demonstrate how the microstructural data arising from the very detailed description of the EBSD data, can provide quantitative conclusions about the mass ration of the melt-rock. All figures are in good quality and the language, I am not a native English speaker, seems to me adequate. This manuscript is appropriate for publication to Solid Earth after minor revision. There are few point that need to be considered: 1. Chapter 5.1 Mineral major and trace elements: a more detailed mineralogical description is needed. 2. Page 7, line 6: why the accelerating voltage vary from 15 to 20 kV? 3. Page 7, line 12: Plagioclase up to 0.8% is high. Explain the origin of this plagioclase 4. Page 10, line 30: Opx with 2.5 wt% CaO is unusual for a mantle xenolith. Probably there are thin cpx exsolution lamellae. Check it. 5. Table 2: Thought there are calculated whole rock major element compositions there is a systematic inverse CaO/Al2O3 ratio. The CaO/Al2O3 ratio in non-metasomatized mantle peridotites is <1. In the case of Table 2, with exception of sample SOU-003, is the ratio CaO/Al2O3>1. Explain why. This is not consistent with reaction with undersaturated melts.

---

## Referee Comment (RC2) · Rachel E. Bernard (Referee) · 30 Mar 2019

This was a well-written and scientifically sound manuscript describing the interaction between deformation and melt along STEP faults. The methods described seem appropriate and the interpretations well-reasoned. I have some comments and suggestions for the authors to consider. That being said, the issues I bring up below are relatively minor and I recommend this paper for acceptance.

Section 5.2: I would have liked to see more of a discussion of the difference in temperatures calculated using the two geothermometers. Is there a geological explanation for this? Should we believe one method over another?

[Figure]

Section 6.1: (page 11, lines 28-30): I was a bit confused by the terminology of "second phase particles," but maybe I'm just unfamiliar with this term. Does it always refer to the phase doing the "pinning"? (Also, there is a misplaced comma in line 29 after the hyphen)

(page 12, lines 8-9): Something is odd about this sentence. As is, it doesn't make sense grammatically, particularly in this part: "relatively cold and then stiffer and high stress..."

Section 6.1.2 (page 13, lines 24-27): This is a more general comment, but is there a reason why the authors chose not to estimate stress magnitude from recrystallized or subgrain size? It seemed odd to me that this was missing, especially because all the authors would have to do is use the already obtained EBSD maps to calculate the average grain size, apply a correction factor, and use a paleopiezometer (e.g., Van der Wal et al. 1993). The authors note how the fine grain sizes of some samples are consistent with high-stress deformation, but don't put a number on this. People might be interested in this value both from a regional deformation perspective but also to address whether or not higher stresses are associated with their [100]-fiber CPO samples vs. orthorhombic CPOs, as is commonly assumed (e.g., Karato et al. 2008, review).

(page 14, lines 16-17): Was there any microstructural evidence (other than weakened CPO) for a switch to diffusion-assisted grain boundary sliding? As an example, were there any 4-grain junctions?

Figure 3: You should specify whether the maps shown are the entire EBSD area mapped for each sample or just a portion of it. If so, if doesn't appear that there would be enough opx or cpx grains in 3B for a reliable bulk CPO.

Figure 9: It is hard to read the yellow text, even with the black outlines.

Figure 10: It is hard to see the tiny volcanoes on the surface. Perhaps just make
them one color (and bigger). It is also difficult to read the text associated with those volcanoes (in red and blue). It might help to have a white box behind each of them.

Overall, again, this was an excellent paper and I hope these comments/suggestions prove to be useful to the authors.

---

## Referee Comment (RC3) · Vasileios Chatzaras (Referee) · 12 Apr 2019

Dear Authors, dear Editor,

This manuscript highlights the microstructural character of shallow subcontinental mantle xenoliths that are interpreted to sample the mantle section of a major Subduction-Transform Edge Propagator (STEP) fault. Microstructural and mineral chemistry data are used to infer synkinematic melt-rock interactions that led to enrichment of refractory harzburgite in coarse-grained clinopyroxene and fine-grained orthopyroxene, at deeper and shallower levels of the system, respectively. The described processes are interpreted to provide insights into melt-present deformation in mantle shear zones,

associated with the operation of STEP faults. The presented data are good and the discussion, interpretations, and conclusions are all consistent with the data presented. The manuscript is well written, and the figures are of high quality. Figure 10 is excellent. Despite my lengthy comments, I find the manuscript very interesting and intriguing, and my recommendation is to be accepted following minor revision. Below, I list some comments and suggestions that the Authors could consider addressing.

1) What are the implications of the described melt-rock interactions on the mechanical behaviour of STEP faults? The sampled suite of xenoliths offers unique insights into processes that take place in the mantle section of a STEP fault, so the Authors could go one step further and explore how the described microtectonic evolution may have affected mantle strength and rheology. For example, the Authors could use the olivine subgrain size, which is mentioned in the Methods section but not included in the manuscript, to determine the stress levels at different depths of the lithosphere. In samples where the microstructures are controlled by dynamic recrystallization, the olivine recrystallized grain size could be used, as well.

2) A long-standing problem in mantle xenolith studies is the lack of a clear foliation and lineation, which leads to the production of thin sections in random orientations relative to the rock shape fabric. As a result, the EBSD-derived crystallographic orientations are rotated so as to match one of the common crystallographic texture types described in the literature. Similar workflow is followed in this study. The main problem here becomes the discrimination between the different orthorhombic CPO patterns. The axial-[100], axial-[010], and orthorhombic symmetries can still be identified (e.g., with the use of the BA-index as done here) without the need to plot the crystallographic texture data relative to the rock shape fabric. As a solution to this problem, the use of X-ray Computed Tomography (XRCT) was recently proposed, where rock fabric can be determined quantitatively by the 3D shape of spinel grains (Chatzaras et al., 2016, already cited in the manuscript). In fact, to the best of my knowledge, the first paper in which XRCT was used for visual determination of the rock fabric in mantle xenoliths,

was of this manuscript's first Author (Hidas et al., 2007). The Authors could use the rock billets from which the thin sections were produced to determine quantitatively the rock fabric using XRCT, and replot the EBSD data relative to the fabric reference framework. That said, here the Authors do not attempt any discrimination between the different orthorhombic CPO patterns, so their workflow is totally appropriate for the level of interpretation. It is just that the use of XRCT would provide information currently inaccessible for the analyzed xenolith suite.

3) Olivine CPO in the coarse-grained xenoliths has a dominant axial-[100] symmetry, while it transitions toward an axial-[010] symmetry in the fine-grained xenoliths, where shearing combined with extensive synkinematic melt-rock interaction is interpreted to take place along a ductile shear zone associated with the Rif-Tell STEP fault. Based on these observations / interpretations, I am thinking of the following CPO and tectonic evolution, which the Authors may want to consider. Olivine axial-[100] CPO symmetry could be the result of constrictional strain associated with mantle upwelling in the slab window beneath the North African margin. An axial-[100] CPO pattern in both olivine and plagioclase was observed in xenoliths from the San Quintin volcanic field in Baja California (van der Werf et al., 2017), which is also interpreted to lie above a slab window (e.g., Zhang et al., 2012), similar to the Oran volcanic field. In the Oran xenoliths, mantle rocks were then captured from the inferred shear zone at the mantle section of the STEP fault. Focused melt migration along the shear zone and potential transpressional deformation (based on Figure 10) may have caused a transition of olivine CPO toward axial-[010] symmetry. The observed variations in microstructures and olivine CPOs could reflect either vertical or lateral heterogeneities in the North African SCLM.

Detailed treatment of either of these comments might require lengthy additions to what is already a reasonable-sized manuscript. These comments should be considered only as suggestions.

Minor comments

Page 1, lines 28-29: The Authors mention that grain size is "uncorrelated with modal variations", while in lines 31-32 (same page), it is mentioned that "Olivine grain size in the fine-grained peridotites depends on the size and volume fraction of the pyroxene grains". How do these statements fit together?

Page 5, lines 11-13: Please state the exact number of samples (and identify their names) in which the thin sections were produced relative to the common structural framework (normal to foliation and parallel to lineation). Also, a suggestion for Figure S1, would be to use the horizontal line and the star (as in Figure 5) to show the foliation and lineation in the samples cut relative to the rock shape fabric.

Page 6, Lines 5-7: I don't think that the Authors present in the manuscript the calculations of the subgrain boundaries length and subgrain density mentioned here. Either remove this description or include the results in the manuscript. Having said that, I think that the manuscript would benefit from the inclusion of these data, if subgrain size is used for estimating differential stress. See comment 1.

Page 6, Lines 8-17: Following on the previous comment, the KAM2, Mis2Mean, and GOS data described here are not presented in the manuscript. Exception is Figure 3, where two Mis2Mean maps are included. The Authors may want to revise the Methods section removing the description of these parameters. Alternatively, they could use the data to describe the microstructural characteristics of different mineral phases and grain sizes.

Page 8, Line 19: Please mention some sample names in which the reader can observe the feature you describe here (elongated patches of clinopyroxene aggregates). It would also be useful to highlight these features in the relevant EBSD phase maps.

Page 8, Line 21: "Strain-free" is an interpretation. Please describe the observations that lead to this interpretation.

Page 8, Line 23: Please highlight on the photomicrograph or EBSD map of Figure 3

these cusp-like terminations at triple junctions.

Page 8, Line 27: "locally showing reaction microstructure" is an interpretation. What are the relevant microscale observations?

Page 9, Lines 16-18: Please be more specific to which samples in Figure 5b you refer. HAM-005b does not show an axial-[010] symmetry.

Page 9, Lines 26-27: Looking the CPO plots, and particularly those of DZ-003, which is the oriented thin section, I am not convinced that this is the case. The maximum of the orthopyroxene [100] axes lies within the foliation plane at high angle to the lineation, although two smaller concentrations near the pole to the foliation are also present. Moreover, please mention which are the oriented thin sections so that the reader can track the information mentioned in the text.

Page 9, Lines 29-30: If orthopyroxene [010] and [001] axes are distributed subparallel to olivine [010] and [001] axes, we would expect the same relationship to hold for the [100] axes, as well. This is not the case in HAM-007, where olivine [100] axes are oriented at high angle to orthopyroxene [100] axes.

Page 10, Lines 4-5: In methods, the Authors describe a 2-12o range for subgrain boundaries, so I am wondering why they chose a different range of angles to analyse low-angle misorientations. Moreover, could the Authors explain the criteria for choosing the 400 $\mu$m grain size threshold for the misorientation analysis? Earlier on (page 8, lines 2-3), they defined the coarse and fine grained porphyroclasts based on a 800 $\mu$m grain size threshold.

Page 11, Lines 16-17: I agree with this statement only for the coarse-grained xenoliths (green color). When it comes to the rest three microstructural types, I do not see a clear trend. I am wondering whether a plot of grain size versus estimated temperature would help the Authors to make their argument more clear. This is quite important point, because if there is no clear positive correlation between grain size and temperature,

the Authors might want to consider the possibility that the xenoliths sample a horizontal strain gradient across the STEP fault. I am also wondering whether any fine-grained xenoliths have been reported from Souahlia. The lack of fine-grained xenoliths might be indicative of an horizontal strain gradient between Souahlia and Ain Temouchent.

Page 11, Lines 25-27: Some more information regarding the calculation of the Zener parameter might be useful to be included in the manuscript. Specifically, were all orthopyroxene and clinopyroxene grains in each sample included in the analysis, or porphyroclasts were excluded? In the latter case, what was the maximum size of grains included? Moreover, I am not sure that we can separate the contribution of orthopyroxene and clinopyroxene grains to the pining of olivine grains. In the current analysis, the underlying assumption is that the only second phase is either orthopyroxene or clinopyroxene, and the rest area/volume is occupied mainly by olivine. Such assumption could work for samples with only a small fraction of the other pyroxene. Otherwise, the two pyroxenes should be considered together.

Page 12, Lines 3-4: In agreement with Figure 8, the Authors state here that olivine grain growth is impeded by the small, interstitial pyroxene grains. However, in page 8, lines 25-26, it is mentioned that in the xenoliths with an equigranular microstructure, the small pyroxene grains "occur in monophase patches rather than showing phase mixing", which is actually not what we see in the cited Figure 3e.

Page 12, Lines 23-25: Development of axial-[100] CPO symmetry in olivine has also been attributed to constrictional strain (Chatzaras et al., 2016).

Page 13, Lines 15-30: I do not think that the one hypothesis necessarily precludes the other. Olivine shearing in the presence of melt could take place in transpressional deformation, where the (001)[100] and (010)[100] (as suggested by the concentration of rotation axes around [001] in Figure 6 for the fine-grained xenoliths) olivine slip systems could both be active due to strain compatibility requirements.

Page 14, Lines 14-15 and 22: Please name the deformation mechanisms.

Papers cited

Van der Werf, T.F., Chatzaras, V., Kriegsman, L., Kronenberg, A., Tikoff, B., Drury, M.R., 2017. Constraints on the rheology of lower crust in a strike-slip plate boundary: Evidence from the San Quintin xenoliths, Baja California, Mexico. Solid Earth, 8, 1211–1239, doi:10.5194/se-2017-45

Zhang, X., Paulssen, H., Lebedev, S., and Meier, T.: 3D shear velocity structure beneath the Gulf of California from Rayleigh wave dispersion, Earth Planet. Sc. Lett., 279, 255–262, 2009.

Sincerely, Vasileios Chatzaras

---

## Author Comment (AC1) · 31 May 2019

**Reviewer #1 (Theodoros Ntaflos)**

**Major and specific comments**

*1. Chapter 5.1 Mineral major and trace elements: a more detailed mineralogical description is needed.*

We think that this chapter contains all the information on the major element composition of the constituent phases that are necessary to reconstruct the melt-rock reaction. We do not have trace element compositions in this manuscript.

*2. Page 7, line 6: why the accelerating voltage vary from 15 to 20 kV?*

The analyses were carried out in the same EPMA lab at the University of Granada in two sessions that were distant from each other in time (in 2012 and 2017). There is no special reason for changing the accelerating voltage between sessions but, during this period, minor modifications in the analytical conditions have been implemented by the EPMA operator in the analysis of geological materials. We applied the routine procedure in each session. Considering that the mineralogical composition and other physical characteristics of the Oran xenoliths are similar, using 15 or 20 kV accelerating voltage does not have any negative impact on the quality of obtained major element data.

*3. Page 7, line 12: Plagioclase up to 0.8% is high. Explain the origin of this plagioclase*

Most Oran mantle xenoliths are plagioclase free, and in those ones where this phase was observed, the modal content typically ranges 0.1-0.2%. There are two samples, HAM-011 and HAM-009, where somewhat higher plagioclase content was detected (0.4 and 0.8%, respectively; Table 1). In these xenoliths, plagioclase is interstitial and usually occurs close to spinel and pyroxenes, typical for dominantly subsolidus formation of plagioclase upon the destabilization of spinel at shallow mantle levels. Observations support the shallow mantle origin of plagioclase-bearing lherzolites from the Oran volcanic field. This transformation reaction, however, is highly dependent on the bulk rock composition (e.g., Borghini et al., 2010, 2011) and, considering the mainly harzburgitic model contents, it is not surprising that the presence of plagioclase is not that common among the studied samples even though possibly some may have also partly equilibrated in shallow mantle levels. We updated the description of plagioclase petrography in the revised manuscript: *"…These occurrences and the observed modal content of plagioclase is nevertheless compatible with dominantly subsolidus origin of plagioclase at the expense of spinel (e.g., Rampone et al., 1993;Borghini et al., 2010 and references therein)…"*

References cited in the answer:

Borghini, G., Fumagalli, P., and Rampone, E.: The Stability of Plagioclase in the Upper Mantle: Subsolidus Experiments on Fertile and Depleted Lherzolite, Journal of Petrology, 51, 229-254, 10.1093/petrology/egp079, 2010.

Borghini, G., Fumagalli, P., and Rampone, E.: The geobarometric significance of plagioclase in mantle peridotites: A link between nature and experiments, Lithos, 126, 42-53, 10.1016/j.lithos.2011.05.012, 2011.

**_4. Page 10, line 30: Opx with 2.5 wt% CaO is unusual for a mantle xenolith. Probably there are thin cpx exsolution lamellae. Check it._**

We appreciate spotting this error in the database. Indeed, large orthopyroxene crystals often contain lamellae of clinopyroxene in their cores (page 8, lines 15-17 in the original manuscript). In this sample, the typical range of CaO in orthopyroxene does not exceed 0.7 wt% and the outlier analysis was taken in the core of orthopyroxene, so we agree with the reviewer that it must correspond to a mixed analysis contaminated by clinopyroxene lamellae. Consequently, we removed this analysis from the database.

**_5. Table 2: Thought there are calculated whole rock major element compositions there is a systematic inverse CaO/Al2O3 ratio. The CaO/Al2O3 ratio in non-metasomatized mantle peridotites is <1. In the case of Table 2, with exception of sample SOU-003, is the ratio CaO/Al2O3>1. Explain why. This is not consistent with reaction with undersaturated melts._**

Most Oran mantle xenoliths —including the coarse-grained peridotites (e.g., page 13, lines 2-3 of the original manuscript)— display petrographic and/or geochemical signatures of melts and various degrees of melt-rock reaction. We propose that the microstructure of the studied mantle xenoliths record both the geochemical evolution of the upwards migrating melt in the SCLM and the lateral heterogeneities in the function of distance to the melt conduits, hence differences of time-integrated melt-rock ratio. The replacement of orthopyroxene by clinopyroxene ±olivine (e.g., presented on the example of xenolith DZ-003 in Fig. 3b) is explained by wehrlitization, which is ascribed to melt-rock reaction either by carbonatitic or undersaturated silicate melts in the literature. Carbonatitic melts result in high whole-rock $CaO/Al_2O_3$, $Na_2O/Al_2O_3$ values, LILE enrichment, and formation of apatite and jadeitic clinopyroxene (e.g., Yaxley et al., 1991).

Although in the Oran mantle xenoliths, the $CaO/Al_2O_3$ ratios are indeed identical to the proposed one for carbonatite metasomatism, the low $Na_2O/Al_2O_3$ (<0.02-0.17; Table 2 vs. expected 0.44-0.49 in Yaxley et al., 1991), the lack of apatite and the low $Na_2O$ content of clinopyroxene (0.92-1.59 wt.%; page 11, line 2 vs. expected 1.5-2.5 wt% in Fig. 2 of Yaxley et al., 1991) rule out that wehrlitization was due to interaction with carbonatites. This is particularly the case of the coarse-grained wehrlite sample SOU-003, where —in addition to the low $Na_2O/Al_2O_3$ ratio (<0.02, Table 2), the low $Na_2O$ content of clinopyroxene (0.8-1.0 wt%; Table S1) and the lack of apatite— the $CaO/Al_2O_3$ is very low, as the Reviewer also pointed out.

Based on the observations we think that the interaction with silicate melt is justified and we favor this model in the manuscript rather than carbonatitic melts. Consequently, the microstructure of the Oran mantle xenoliths can be explained by the evolution of the silicate melt from primitive to evolved compositions during upward migration in the lithospheric mantle column, which was presented in the manuscript with sufficient details in page 15, line 17 onward in the original version. We nevertheless added more details to this chapter in the revised manuscript to exclude the interaction with carbonatite melts as: *"…Unlike the expected high $Na_2O/Al_2O_3$ values accompanied with the presence of apatite and jadeitic clinopyroxene in wehrlites related to carbonatites ($Na_2O/Al_2O_3$ in whole rock: 0.44-0.49, $Na_2O$ in Cpx: 1.5-2.5*

*wt.%; Yaxley et al., 1991; Raffone et al., 2009), the Oran xenoliths have petrographic and geochemical signatures ($Na_2O/Al_2O_3$ in whole rock: <0.02-0.17; $Na_2O$ in Cpx: 0.92-1.59 wt.%; Table 2, Table S1) typical of mantle wall rocks that interacted with melts relatively close to magma conduits or porous flow channels (Bodinier et al., 1990;Ionov et al., 2005)…"*

Reference cited in the answer:

Yaxley, G. M., Crawford, A. J., and Green, D. H.: Evidence for carbonatite metasomatism in spinel peridotite xenoliths from western Victoria, Australia., Earth and Planetary Science Letters, 107, 305-317, 1991.

---

## Author Comment (AC2) · 31 May 2019

**Reviewer #2 (Rachel E. Bernard)**

**Major and specific comments**

***Section 5.2: I would have liked to see more of a discussion of the difference in temperatures calculated using the two geothermometers. Is there a geological explanation for this? Should we believe one method over another?***

In the manuscript, we used two geothermometeric methods: the Opx-Cpx equilibrium thermometer of Brey and Köhler (1990) and the empirical Opx thermometer of Witt-Eickschen and Seck (1991). The decoupling between the different geothermometric results is often observed in mantle rocks that record various degrees of disequilibrium. This disequilibrium is typically kinetic due to different diffusion of Ca and Al in pyroxenes.

In the Oran mantle xenoliths, clinopyroxene and orthopyroxene major element compositions provide a relatively wide range of temperatures along with decoupling between the results of different thermometers. This is observed less frequently in xenoliths than in massif peridotites. We consider that the decoupling is due to the melt-rock reaction, which resulted in disequilibrium of pyroxenes in some xenoliths. Although the disequilibrium involves uncertainties in the calculated temperatures in these samples, this is the only way to estimate the potential range of temperature in the rocks. Therefore, we think that the detected differences do not reflect the reliability of one thermometer over the other but, rather, highlight that the chemical equilibrium has been perturbed during the melt-rock reaction. We added a brief explanation to the Geothermometry chapter but a more detailed discussion is beyond the scope of the manuscript: *"…We observe a weak match of the two geothermometric methods only in a few coarse-grained rocks, typically in the range of ca. 920-1050 °C, which potentially reflects chemical disequilibrium between the pyroxenes…"*

References cited in the answer:

Brey, G. P., and Köhler, T.: Geothermobarometry in four-phase lherzolites. II. New thermobarometers, and practical assessment of existing thermobarometers, Journal of Petrology, 31, 1353-1378, 1990.
Witt-Eickschen, G., and Seck, H. A.: Solubility of Ca and Al in orthopyroxene from spinel peridotite - an improved version of an empirical geothermometer, Contributions to Mineralogy and Petrology, 106, 431-439, 1991.

***Section 6.1: (page 11, lines 28-30): I was a bit confused by the terminology of "second phase particles," but maybe I'm just unfamiliar with this term. Does it always refer to the phase doing the "pinning"? (Also, there is a misplaced comma in line 29 after the hyphen)***

Second phase particle is a terminology of material sciences background and, to our best knowledge, its use is also common in Earth Sciences. The term is often associated with the pinning effect of these textural elements but it is not exclusively used in this context everywhere. In fact, a second phase particle can refer to any secondary phases that are present in a polymineralic assemblage but the pinning effect largely depends on the size and distribution of

such phases in the texture (grain boundaries vs. inclusions, crystal aggregates vs. dispersed crystals in the fabric). This is the analysis that we carried out in Fig. 8 of the manuscript.

We deleted the comma in line 29.

***(page 12, lines 8-9): Something is odd about this sentence. As is, it doesn't make sense grammatically, particularly in this part: "relatively cold and then stiffer and high stress: : :"***

We clarified the sentence in the revised version: *"…Grain size reduction in mantle xenoliths is symptomatic of increasing strain localization in relatively cold, stiff and high stress domains of the mantle lithosphere (e.g., Drury et al., 1991;Karato, 2008;Linckens et al., 2011b;Vauchez et al., 2012;Tommasi and Vauchez, 2015)…"*

***Section 6.1.2 (page 13, lines 24-27): This is a more general comment, but is there a reason why the authors chose not to estimate stress magnitude from recrystallized or subgrain size? It seemed odd to me that this was missing, especially because all the authors would have to do is use the already obtained EBSD maps to calculate the average grain size, apply a correction factor, and use a paleopiezometer (e.g., Van der Wal et al. 1993). The authors note how the fine grain sizes of some samples are consistent with high-stress deformation, but don't put a number on this. People might be interested in this value both from a regional deformation perspective but also to address whether or not higher stresses are associated with their [100]-fiber CPO samples vs. orthorhombic CPOs, as is commonly assumed (e.g., Karato et al. 2008, review).***

The main reason for not including paleopiezometry in the manuscript is that this method only works for dynamically recrystallized grain sizes. Considering grains other than those of dynamic recrystallization origin would certainly return erroneous stress values in the paleopiezometric estimations. In the Oran mantle xenoliths, olivine and pyroxene grain sizes are largely influenced by melt-rock reaction, and we have no tools to differentiate dynamically recrystallized grains from those produced during the melt-rock reaction. This issue is critical in the fine-grained rocks but cannot be disregarded in the coarse-grained xenoliths either, where at least a fraction of small orthopyroxene is of reactional origin. Consequently, we decided not to use paleopiezometry in the revised manuscript either.

*(page 14, lines 16-17): Was there any microstructural evidence (other than weakened CPO) for a switch to diffusion-assisted grain boundary sliding? As an example, were there any 4-grain junctions?*

The switch in the deformation mechanism is supported by the observations listed in the paragraph and we have not found any further microstructural evidence for the presence of melts.

*Figure 3: You should specify whether the maps shown are the entire EBSD area mapped for each sample or just a portion of it. If so, if doesn't appear that there would be enough opx or cpx grains in 3B for a reliable bulk CPO.*

The images have been cropped for aesthetic purposes in the figure but the data behind the analyses are considered for the entire analyzed area. We have clarified this in the revised figure caption: *"…Note that images in the figure are cropped out from EBSD maps; see supplementary Fig. S1 for the entire maps…"*

*Figure 9: It is hard to read the yellow text, even with the black outlines.*

We updated the figure to enhance legibility:

[Figure]

*Figure 10: It is hard to see the tiny volcanoes on the surface. Perhaps just make them one color (and bigger). It is also difficult to read the text associated with those volcanoes (in red and blue). It might help to have a white box behind each of them.*

We increased the size of the volcanoes and we added a white box in the background, but the blue and red colors reflect geochemistry of the igneous rocks, thus we consider this information important.

---

## Author Comment (AC3) · 31 May 2019

**Reviewer #3 (Vasileios Chatzaras)**

**Major comments**

*1) What are the implications of the described melt-rock interactions on the mechanical behaviour of STEP faults? The sampled suite of xenoliths offers unique insights into processes that take place in the mantle section of a STEP fault, so the Authors could go one step further and explore how the described microtectonic evolution may have affected mantle strength and rheology. For example, the Authors could use the olivine subgrain size, which is mentioned in the Methods section but not included in the manuscript, to determine the stress levels at different depths of the lithosphere. In samples where the microstructures are controlled by dynamic recrystallization, the olivine recrystallized grain size could be used, as well.*

The analysis of the olivine subgrain size provided ambiguous results, therefore, we decided not to include it in the manuscript and the database. The reference to this data in the methodology chapter is a mistake but we removed it from the revised text. As for the piezometry based on the recrystallized grain size, we refer to our answer to Reviewer #2.

*2) A long-standing problem in mantle xenolith studies is the lack of a clear foliation and lineation, which leads to the production of thin sections in random orientations relative to the rock shape fabric. As a result, the EBSD-derived crystallographic orientations are rotated so as to match one of the common crystallographic texture types described in the literature. Similar workflow is followed in this study. The main problem here becomes the discrimination between the different orthorhombic CPO patterns. The axial-[100], axial-[010], and orthorhombic symmetries can still be identified (e.g., with the use of the BA-index as done here) without the need to plot the crystallographic texture data relative to the rock shape fabric. As a solution to this problem, the use of X-ray Computed Tomography (XRCT) was recently proposed, where rock fabric can be determined quantitatively by the 3D shape of spinel grains (Chatzaras et al., 2016, already cited in the manuscript). In fact, to the best of my knowledge, the first paper in which XRCT was used for visual determination of the rock fabric in mantle xenoliths, was of this manuscript's first Author (Hidas et al., 2007). The Authors could use the rock billets from which the thin sections were produced to determine quantitatively the rock fabric using XRCT, and replot the EBSD data relative to the fabric reference framework. That said, here the Authors do not attempt any discrimination between the different orthorhombic CPO patterns, so their workflow is totally appropriate for the level of interpretation. It is just that the use of XRCT would provide information currently inaccessible for the analyzed xenolith suite.*

We agree with the reviewer that micro-CT may provide crucial information for microstructural studies. Unfortunately, at the time when the thin sections on the Oran mantle xenoliths were made, we did not have access to micro-CT and the remaining rock chips are now too small for such analyses. Consequently, we can only use the traditional methods from the literature to differentiate between CPO patterns.

*3) Olivine CPO in the coarse-grained xenoliths has a dominant axial-[100] symmetry, while it transitions toward an axial-[010] symmetry in the fine-grained xenoliths, where shearing combined with extensive synkinematic melt-rock interaction is interpreted to take place along a ductile shear zone associated with the Rif-Tell STEP fault. Based on these observations / interpretations, I am thinking of the following CPO and tectonic evolution, which the Authors may want to consider. Olivine axial-[100] CPO symmetry could be the result of constrictional strain associated with mantle upwelling in the slab window beneath the North African margin. An axial-[100] CPO pattern in both olivine and plagioclase was observed in xenoliths from the San Quintin volcanic field in Baja California (van der Werf et al., 2017), which is also interpreted to lie above a slab window (e.g., Zhang et al., 2012), similar to the Oran volcanic field. In the Oran xenoliths, mantle rocks were then captured from the inferred shear zone at the mantle section of the STEP fault. Focused melt migration along the shear zone and potential transpressional deformation (based on Figure 10) may have caused a transition of olivine CPO toward axial-[010] symmetry. The observed variations in microstructures and olivine CPOs could reflect either vertical or lateral heterogeneities in the North African SCLM.*

We see no contradiction between our conceptual model on the development of olivine CPO in a melt-lubricated STEP mantle shear zone and the proposition outlined by the Reviewer. We agree with his points and we added more details on the conceptual model to the discussion (chapters 6.1.2 and 6.3) following the Reviewer's suggestions.

Chapter 6.1.2:

*"...Thus, we favor the latter hypothesis that is shear deformation of olivine in the presence of melt to explain the shift towards [010]-fiber olivine patterns in the equigranular peridotites. Alternatively, the transition of olivine CPO symmetry toward axial-[010] in the fine-grained xenoliths may have developed enhanced by several of the above factors, of which the most likely scenario would be focused melt migration in a transpressional mantle shear zone, considering the geodynamic environment of the Oran volcanic field. Nevertheless, in the observed textures, the crystallographic axes of small orthopyroxenes are weakly oriented and, occasionally, distributed subparallel to those of olivine (Fig. 5b; Fig. S1)."*

Chapter 6.3:

*"...This deformation may either be an older event preserved in the rocks, or correspond to the lateral mantle flow from the Canary plume and mantle upwelling beneath the N-African margin, resulting in axial-[100] olivine CPO symmetry that developed due to constrictional strain. [...] These results suggest that wehrlitization of the Oran SCLM is a relatively young event, which took place in late Miocene to early Pleistocene times, and the observed variations in microstructures and olivine CPOs could reflect vertical and/or lateral heterogeneities along the N-African margin..."*

**Minor comments**

*Page 1, lines 28-29: The Authors mention that grain size is "uncorrelated with modal variations", while in lines 31-32 (same page), it is mentioned that "Olivine grain size in the fine-grained peridotites depends on the size and volume fraction of the pyroxene grains". How do these statements fit together?*

The grain size is uncorrelated with modal variations in the sense that it does not correlate to lithological classification, i.e., there are coarse-grained and fine-grained harzburgite or wehrlite among the studied xenoliths. Rather than lithology, the dependence of the olivine grain size on the modal content of pyroxene is clear if not only the volume fraction but the size of these second phase particles is considered as well (cf. Fig. 8). We clarified the apparent contradiction in the revised manuscript:

*"…The microstructures of mantle xenoliths show a variable grain size ranging from coarse granular to fine-grained equigranular textures uncorrelated with lithology…"*

***Page 5, lines 11-13: Please state the exact number of samples (and identify their names) in which the thin sections were produced relative to the common structural framework (normal to foliation and parallel to lineation). Also, a suggestion for Figure S1, would be to use the horizontal line and the star (as in Figure 5) to show the foliation and lineation in the samples cut relative to the rock shape fabric.***

We now indicate the oriented thin sections with an asterisk in the first column of Table 1 in the revised manuscript and we updated the supplementary Fig. S1 with the information requested by the Reviewer.

[revised manuscript text omitted]

***Page 6, Lines 5-7: I don't think that the Authors present in the manuscript the calculations of the subgrain boundaries length and subgrain density mentioned here. Either remove this description or include the results in the manuscript. Having said that, I think that the manuscript would benefit from the inclusion of these data if subgrain size is used for estimating differential stress. See comment 1.***

During the compilation of the manuscript, we attempted to calculate the subgrain size and subgrain densities from EBSD data but the results are highly influenced by analytical artifacts (e.g., charging in some critical xenoliths) and are also moderately dependent on the step size of the EBSD maps. Consequently, we decided not to use this ambiguous data in the final version

but the corresponding description has been accidentally left in the methods chapter. We removed this part from the revised version.

As for the estimation of differential stress based on recrystallized grain size, we refer to our answer to the related comment of Reviewer #2.

***Page 6, Lines 8-17: Following on the previous comment, the KAM2, Mis2Mean, and GOS data described here are not presented in the manuscript. Exception is Figure 3, where two Mis2Mean maps are included. The Authors may want to revise the Methods section removing the description of these parameters. Alternatively, they could use the data to describe the microstructural characteristics of different mineral phases and grain sizes.***

Discussing all the calculated parameters is beyond the scope of the research and, in the manuscript, we address only the most important ones. We nevertheless include the results of the calculations in a database, which is provided as supplementary material. In the revised version, we refer to the corresponding table:*"... For the complete database of calculated parameters, see Table S1. .."*

***Page 8, Line 19: Please mention some sample names in which the reader can observe the feature you describe here (elongated patches of clinopyroxene aggregates). It would also be useful to highlight these features in the relevant EBSD phase maps.***

We refer to some characteristic samples in the revised manuscript: *"...Clinopyroxene may also form large crystals that are not porphyroclasts but aggregates of several millimetric crystals in elongated patches in the plane of the foliation (e.g., xenoliths GU-001, HAM-001, HAM-009; Fig. S1)..."*

***Page 8, Line 21: "Strain-free" is an interpretation. Please describe the observations that lead to this interpretation.***

We added the observations that led to the interpretation: *"...Large pyroxenes are rarely observed in the equigranular texture. On the other hand, both pyroxenes occur as a fine-grained, elongated (with breadth <100 microns) mineral fraction free from petrographic signs of intracrystalline lattice distortion and usually dispersed in the rocks without showing textural preference to porphyroclasts..."*

***Page 8, Line 23: Please highlight on the photomicrograph or EBSD map of Figure 3 these cusp-like terminations at triple junctions.***

We indicate representative examples of irregularly shaped small pyroxenes that are distributed at olivine-olivine grain boundaries, or have cusp-like termination at triple junctions in the revised Fig. 3e:

[Figure]

***Page 8, Line 27: "locally showing reaction microstructure" is an interpretation. What are the relevant microscale observations?***

We clarified the description in the revised manuscript: *"...In some clinopyroxene-rich depleted lherzolite (e.g., DZ-003), the fine-grained clinopyroxene occur as a rim on orthopyroxene porphyroclasts with irregular phase boundaries and identical crystallographic orientation of the two pyroxenes (Fig. 3b)..."*

***Page 9, Lines 16-18: Please be more specific to which samples in Figure 5b you refer. HAM-005b does not show an axial-[010] symmetry.***

We refer to a representative sample in the revised manuscript: *"...In the orthorhombic CPO symmetries that show tendency towards [010]-fiber pattern, typical for some fine-grained xenoliths, we observe a strong maximum of [010] perpendicular to the foliation and a very weak girdle of [100] and [001] axes in the plane of the foliation (e.g., sample HAM-007 in Fig. 5b)..."*

***Page 9, Lines 26-27: Looking the CPO plots, and particularly those of DZ-003, which is the oriented thin section, I am not convinced that this is the case. The maximum of the orthopyroxene [100] axes lies within the foliation plane at high angle to the lineation, although two smaller concentrations near the pole to the foliation are also present. Moreover, please mention which are the oriented thin sections so that the reader can track the information mentioned in the text.***

We agree with the reviewer that there are two point-like maxima of [100]-axes in xenolith DZ-003. The apparently stronger maximum at a high angle to the lineation in the plane of the foliation, however, belongs to a few (<5) coarse orthopyroxene crystals and their remnants that have been left behind in their surroundings after the original porphyroclast was reacted to form secondary clinopyroxene ±olivine (Fig. R1). This fragmentation results in anomalously high-density contours in the pole figures related to these few, similarly oriented crystal aggregates that otherwise represent only a minority of the orthopyroxene population. This issue is typical for coarse-grained microstructures where the number of grains of a given mineral phase is limited, and even a minor over-segmentation can result in anomalously high contour densities. In our case, the increased number of similarly oriented orthopyroxene crystals is due to the melt-rock reaction that left behind some small orthopyroxene crystals from the original porphyroclast (e.g., inset in Fig. R1). We nevertheless did not want to delete any data points or artificially modify the dataset but we kept this artifact in mind when presenting the CPO data of this sample in the text. The second maximum in Fig. 5 of the manuscript, which has the [100]-axes distributed at the pole of the foliation, comes from various grains and supports our description provided in page 9, lines 26-27 of the original manuscript.

[Figure]

***Fig. R1:*** *Phase map of xenolith DZ-003 highlighting only those orthopyroxene crystals (in vivid blue) that have anomalous orientations Fig. 5 of the manuscript. In the pole figure only the orientation of these crystals are shown. Inset highlights a coarse orthopyroxene crystal with its remnants left behind after orthopyroxene-consuming melt-rock reaction. For the full dataset, see Fig. 5 and Fig. S1 of the manuscript.*

***Page 9, Lines 29-30: If orthopyroxene [010] and [001] axes are distributed subparallel to olivine [010] and [001] axes, we would expect the same relationship to hold for the [100] axes, as well. This is not the case in HAM-007, where olivine [100] axes are oriented at high angle to orthopyroxene [100] axes.***

The subparallel distribution between olivine and orthopyroxene crystallographic axes is the strongest in the [010] axes, while the other two main crystallographic axes are more dispersed. We do not know the exact mechanism that results in such a crystallographic relationship between olivine and orthopyroxene but the observation is not new in the literature. We agree with the reviewer and we clarified the description in the revised manuscript: *"…In the fine-grained rocks —particularly in the equigranular samples— the main crystallographic axes of orthopyroxene are generally distributed subparallel to those of olivine (e.g., [010]$_{Ol}$ || [010]$_{Opx}$ in xenoliths HAM-007 and HAM-018 in Fig. 5b). Such subparallel distribution is characteristic mostly for [010]- and, less frequently, [100]- or [001]-axes…"*

***Page 10, Lines 4-5: In methods, the Authors describe a 2-12° range for subgrain boundaries, so I am wondering why they chose a different range of angles to analyse low-angle misorientations. Moreover, could the Authors explain the criteria for choosing the 400 μm grain size threshold for the misorientation analysis? Earlier on (page 8, lines 2-3), they defined the coarse and fine grained porphyroclasts based on a 800 μm grain size threshold.***

The difference in the range for defining subgrain boundaries is an error, the correct one is indeed 2-12° and we applied these values during data analysis. We corrected the range in the revised version.

Both threshold values are empirical and serve to identify features of interest in a given analysis. For the textural classification, the ca. 800 μm threshold (expressed as area-weighted average grain size; Fig. 2) in olivine worked the best to differentiate the fine-grained and coarse-grained xenolith textures from each other (page 8, lines 2-3). Note that the area-weighted average grain size is a single value per mineral phase in a given xenolith. However, in the analysis of rotation axes accommodating low-angle misorientations, the emphasis was put rather on the pyroxenes than on olivine, considering that in olivine the rotation axes show <0vw> orientations irrespective of the grain size (page 10, lines 6-8 of the original manuscript). For this purposes, the 400 μm threshold (expressed as equivalent diameter) allowed us to correctly differentiate the two distinct populations of pyroxenes presented in the petrography, and to carry out their detailed analyses. Note that in the petrographic description of the fine-grained pyroxenes the <100 microns diameter (page 8, line 20 of the original manuscript) refers to the breadth of crystals, whereas in the analysis of rotation axes the threshold corresponds to their equivalent diameter. As stated before, in case of olivine our choice of empirical value has no impact on the results and we made the nature of the threshold values clear in the revised manuscript: *"...Based on the average grain size of olivine in these transitional textures, we distinguish coarse-grained porphyroclastic (olivine >0.8 mm area-weighted average grain size; 8 samples, 32%) and fine-grained porphyroclastic texture types (olivine <0.8 mm area-weighted average grain size; 2 samples, 8%) (Fig. 2b). […] On the other hand, both pyroxenes occur as a fine-grained, elongated (with breadth <100 microns) mineral fraction free from petrographic signs of intracrystalline lattice distortion and usually dispersed in the rocks without showing textural preference to porphyroclasts…"*

***Page 11, Lines 16-17: I agree with this statement only for the coarse-grained xenoliths (green color). When it comes to the rest three microstructural types, I do not see a clear trend. I am wondering whether a plot of grain size versus estimated temperature would help the Authors to make their argument more clear. This is quite important point, because if there is no clear positive correlation between grain size and temperature, the Authors might want to consider the possibility that the xenoliths sample a horizontal strain gradient across the STEP fault. I am also wondering whether any fine-grained xenoliths have been reported from Souahlia. The lack of fine-grained xenoliths might be indicative of an horizontal strain gradient between Souahlia and Ain Temouchent.***

As Fig. 2b reflects, there is a homogenization of grain size from the coarse-grained to the fine-grained textures. The textural classification of the xenoliths is based on this figure, thus the grouping of samples by texture in Fig. 7b already contains correlation of the calculated temperatures with olivine grain size. Nevertheless, Fig. 2b also reflects that, particularly in the

porphyroclastic samples, significant differences may exist between the grain size of olivine and pyroxenes. Therefore, we do not know to which grain size the temperature should be compared in a diagram proposed by the reviewer (maximum/average porphyroclast, median, or average small grains, to name a few possibilities). We are also wondering on the expected trend on this diagram that could unambiguously support/eliminate the existence of horizontal strain gradients in the Oran SCLM. We believe that such a plot would not provide information on this question and, consequently, we have not changed the corresponding figure.

Finally, we have not found fine-grained xenoliths in Souahlia but mantle xenoliths are less frequent in this outcrop than elsewhere in Oran, therefore we prefer not to conclude that this texture type is not present in the mantle beneath the locality.

***Page 11, Lines 25-27: Some more information regarding the calculation of the Zener parameter might be useful to be included in the manuscript. Specifically, were all orthopyroxene and clinopyroxene grains in each sample included in the analysis, or porphyroclasts were excluded? In the latter case, what was the maximum size of grains included? Moreover, I am not sure that we can separate the contribution of orthopyroxene and clinopyroxene grains to the pining of olivine grains. In the current analysis, the underlying assumption is that the only second phase is either orthopyroxene or clinopyroxene, and the rest area/volume is occupied mainly by olivine. Such assumption could work for samples with only a small fraction of the other pyroxene. Otherwise, the two pyroxenes should be considered together.***

In fact, the idea with the analysis presented in Fig. 8 is to see if the olivine grain size was controlled by the presence of second phase particles due to pinning. For the sake of simplicity, we consider only pyroxenes in this analysis, as the total amount of spinel, plagioclase and amphibole does not exceed a few modal percents in the studied xenoliths and their mixing with olivine is negligible compared to the pyroxenes. In the analysis, we intentionally decided to address the role of pyroxenes separately and no grains have been left out from the calculations. The main message of these diagrams is that (1) pyroxenes exerted pinning on the olivine grain size particularly in the fine-grained xenoliths, and (2) the two pyroxenes might have a different effect in the coarse-grained textures. Evidently, if both pyroxenes were presented together in a single diagram, the overall distribution would not change but it would be impossible to recognize this subtle difference between the behavior of clinopyroxene and orthopyroxene in the coarse-grained xenoliths. We believe that the current version of the analyses provides more insights into the pinning effect and, consequently, we decided not to merge the clinopyroxenes and orthopyroxenes in a single pyroxene database.

***Page 12, Lines 3-4: In agreement with Figure 8, the Authors state here that olivine grain growth is impeded by the small, interstitial pyroxene grains. However, in page 8, lines 25-26, it is mentioned that in the xenoliths with an equigranular microstructure, the small pyroxene grains "occur in monophase patches rather than showing phase mixing", which is actually not what we see in the cited Figure 3e.***

There is no contradiction, but we must admit that our choice of word is confusing in the cited paragraph (page 8, lines 25-26). "Monophase patches" intended to describe microstructural domains that contain only one type of pyroxene intermixed with olivine, i.e., small clino- and

orthopyroxene rarely occur intermixed with each other. As the reviewer pointed out, Fig. 3 indeed shows that olivine is often intermixed either with orthopyroxene or with clinopyroxene in these patches, but only rarely with the two mineral phases at the same time in the same domain. This observation is also consistent with the analysis of second phase particles presented in Fig. 8, particularly in the fine-grained xenoliths. We clarified the petrographic description in the revised manuscript: *"…Small orthopyroxene and clinopyroxene are generally present with a proportion ranging from 80:20 to 60:40 in a given xenolith, respectively, and they typically occur in patches rich in small olivines that are heterogeneously intermixed with either clino- or orthopyroxene (Fig. 3e). Phase mixing between the two pyroxenes is rare in such domains…"*

***Page 12, Lines 23-25: Development of axial-[100] CPO symmetry in olivine has also been attributed to constrictional strain (Chatzaras et al., 2016).***

We included this reference in the revised paragraph.

***Page 13, Lines 15-30: I do not think that the one hypothesis necessarily precludes the other. Olivine shearing in the presence of melt could take place in transpressional deformation, where the (001)[100] and (010)[100] (as suggested by the concentration of rotation axes around [001] in Figure 6 for the fine-grained xenoliths) olivine slip systems could both be active due to strain compatibility requirements.***

We added this alternative to the text: *"…Alternatively, the transition of olivine CPO symmetry toward axial-[010] in the fine-grained xenoliths may have developed enhanced by several of the above factors, of which the most likely scenario would be focused melt migration in a transpressional mantle shear zone, considering the geodynamic environment of the Oran volcanic field…"*

***Page 14, Lines 14-15 and 22: Please name the deformation mechanisms.***

We clarified the sentence in the revised version: *"…These observations attest for the increasing impact of melts on the deformation from coarse-grained to fine-grained microstructures…"*